# The cuticular hydrocarbon profiles of honey bee workers develop via a socially-modulated innate process

Cassondra L Vernier[1], Joshua J Krupp[2], Katelyn Marcus[1], Abraham Hefetz[3], Joel D Levine[2], Yehuda Ben-Shahar[1]*

[1]Department of Biology, Washington University in Saint Louis, Saint Louis, United States; [2]Department of Biology, University of Toronto Mississauga, Mississauga, Canada; [3]Department of Zoology, Tel Aviv University, Tel Aviv, Israel

**Abstract** Large social insect colonies exhibit a remarkable ability for recognizing group members via colony-specific cuticular pheromonal signatures. Previous work suggested that in some ant species, colony-specific pheromonal profiles are generated through a mechanism involving the transfer and homogenization of cuticular hydrocarbons (CHCs) across members of the colony. However, how colony-specific chemical profiles are generated in other social insect clades remains mostly unknown. Here we show that in the honey bee (*Apis mellifera*), the colony-specific CHC profile completes its maturation in foragers via a sequence of stereotypic age-dependent quantitative and qualitative chemical transitions, which are driven by environmentally-sensitive intrinsic biosynthetic pathways. Therefore, the CHC profiles of individual honey bees are not likely produced through homogenization and transfer mechanisms, but instead mature in association with age-dependent division of labor. Furthermore, non-nestmate rejection behaviors seem to be contextually restricted to behavioral interactions between entering foragers and guards at the hive entrance.

DOI: https://doi.org/10.7554/eLife.41855.001

*For correspondence:
benshahary@wustl.edu

Competing interests: The authors declare that no competing interests exist.

## Introduction

The ability to recognize 'self' plays an important role in regulating diverse processes across biological organizational levels (*Tsutsui, 2004*). Analogous to the acquired immunity system, which depends on self-recognition at the cellular and molecular levels (*Boehm, 2006*), adaptive organismal social interactions often depend on the recognition of kin and/or group-members to increase cooperation or to suppress inbreeding (*Hamilton, 1964a*; *Hamilton, 1964b*; *Pusey and Wolf, 1996*; *Trivers, 1971*; *West et al., 2007*; *Wilkinson, 1988*). One remarkable example of organismal recognition of 'self' comes from colonies of social insects, which depend on a robust non-nestmate discrimination system (more commonly called 'nestmate recognition') to prevent the loss of expensive resources to non-nestmates, and to maintain overall colony integrity (*Hefetz, 2007*; *van Zweden and D'Ettorre, 2010*).

As in other self-recognition systems, theoretical models suggest that nestmate recognition in social insect colonies depends on the ability of individual colony members to reliably match colony-specific phenotypic cues, or 'labels', carried by other colony members, to stored neural 'templates' (*Buckle and Greenberg, 1981*; *Errard, 1994*; *Gamboa et al., 1986*; *Getz, 1982*; *Hölldobler and Michener, 1980*; *Lacy and Sherman, 1983*; *Reeve, 1989*; *Tsutsui, 2004*; *van Zweden and D'Ettorre, 2010*). In some social insect species, the cues used in recognizing individual members of the colony have been reported to be visual (*Baracchi et al., 2015*), but in most cases are thought to be chemical (*van Zweden and D'Ettorre, 2010*). Cuticular hydrocarbons (CHCs), which evolved to

**eLife digest** Honey bees are social insects that live in large groups called colonies, within structures known as hives. The young adult bees stay within the hive to build nests and care for the young, while the older bees leave the hive to forage for food. Honey bees store food and other valuable resources in their hives, so they are often targeted by predators, parasites and 'robber' bees from other colonies. Therefore, it is important for bees to determine whether individuals trying to enter the nest are group members or intruders.

While it is known that social insects use blends of waxy chemicals called cuticular hydrocarbons to identify group members at the entrance to the colony, it is not clear how members of the same colony acquire a similar blend of cuticular hydrocarbons. Some previous work suggested that in some ant species (which are also social insects), colony members exchange cuticular hydrocarbons with each other so that all members of the colony are covered with a similar blend of chemicals. However, it was not known whether honey bees also share cuticular hydrocarbons between colony members in order to identify members of a hive.

Vernier et al. used chemical, molecular and behavioral approaches to study the cuticular hydrocarbons found on honey bees. The results show that, rather than exchanging chemicals with other members of their colony, individual bees make their own blends of cuticular hydrocarbons. As a bee ages it makes different blends of cuticular hydrocarbons, and by the time it starts to leave the hive to forage it makes a blend that is specific to the colony it belongs to. The production of this final blend is influenced by the environment within the hive.

Thus, the findings of Vernier et al. indicate that honey bees guarding the entrance to a hive can only identify non-colony-member forager bees as intruders, rather than any non-colony-member bee that happens upon the hive entrance. Honey bees play an essential role in pollinating many crop plants so understanding how these insects maintain their social groups may help to improve agriculture in the future. Furthermore, this work may aid our understanding of how other social insects interact in a variety of biological situations.

DOI: https://doi.org/10.7554/eLife.41855.002

function as hydrophobic, anti-desiccant barriers in terrestrial arthropods, have been co-opted to also function as pheromones in diverse insect communication systems, including nestmate recognition in social insect species (*Chung and Carroll, 2015*; *van Zweden and D'Ettorre, 2010*). Whether the overall profile, or more specific components of it, represent the actual nestmate recognition cue remains unknown. However, previous studies have indicated that variations in the relative amounts of each compound in the CHC profile across individuals from different colonies are likely sufficient for the chemical recognition of nest membership (*van Zweden and D'Ettorre, 2010*). Nevertheless, how large groups of hundreds to thousands of individuals coordinate the production and recognition of a robust colony-specific chemical cue remains unknown for most species.

Because members of social insect colonies are often genetically related, it was initially assumed that the production of similar colony-specific pheromones by individual colony members is intrinsically driven by shared allelic variants (*Crozier and Dix, 1979*; *Getz, 1982*; *Getz, 1981*). However, empirical studies revealed that, surprisingly, in many social insect species colony and social environmental factors play the most dominant role in defining colony-specific cues, and can often mask genetic relatedness (*Breed et al., 1988*; *Downs and Ratnieks, 1999*; *Heinze et al., 1996*; *Lahav et al., 2001*; *Liang and Silverman, 2000*; *Singer and Espelie, 1996*; *Stuart, 1988*). Although these colony 'environmental' factors remain unknown for most social insect species, it has been suggested that contributions from nest building materials (*Breed et al., 1988*; *Couvillon et al., 2007*; *D'ettorre et al., 2006*; *Espelie et al., 1990*; *Singer and Espelie, 1996*), the queen (*Carlin and Hölldobler, 1988*; *Carlin and Holldobler, 1987*; *Carlin and Holldobler, 1986*; *Carlin and Hölldobler, 1983*), and diet (*Buczkowski et al., 2005*; *Buczkowski and Silverman, 2006*; *Liang and Silverman, 2000*; *Richard et al., 2004*; *Richard et al., 2007*) could, at least in part, provide unique chemical components to the chemical signature shared by colony members. Consequently, empirical and theoretical studies suggested that individual colony members acquire their colony-specific chemical signature largely through a homogenization process involving the exchange of relevant chemicals,

including CHCs, through interactions between colony members or contact with nest building materials, often referred to as the 'Gestalt' model (*Crozier and Dix, 1979*). Empirical evidence in support of this model has been reported for a few ant species, which are known to transfer mixed blends of CHCs between individuals through trophallaxis and grooming via the action of the postpharyngeal gland (PPG) (*Boulay et al., 2000*; *Lenoir et al., 2001*; *Meskali et al., 1995*; *Soroker et al., 1994*; *Soroker et al., 1995b*; *van Zweden et al., 2010*). However, other studies suggest that such CHC homogenization processes might not fully represent how colony-specific chemical cues develop in all social insect species. For example, some ant species do not display robust trophallaxis behaviors, the main mode of chemical transfer across colony members (*Soroker et al., 1994*; *Soroker et al., 1995b*), and in others, the CHC profiles of individual colony members are likely modulated by genetic relatedness (*Teseo et al., 2014*), age (*Cuvillier-Hot et al., 2001*; *Teseo et al., 2014*), and/or task (*Martin and Drijfhout, 2009*; *Sturgis and Gordon, 2013*; *Wagner et al., 2001*; *Wagner et al., 1998*). Together, these data suggest that the regulation of chemical cues in different species is more variable and complex than initially hypothesized (*Esponda et al., 2015*; *Newey, 2011*; *Sturgis and Gordon, 2012*), and remains unknown for most social insect species.

Consequently, here we investigated the development of CHC profiles and nestmate recognition cues in the European honey bee, *Apis mellifera*, a species of economic importance and one of the best studied social insect species. Numerous previous studies have demonstrated that honey bees exhibit a robust nestmate recognition system that is based on the chemical recognition of pheromones (*van Zweden and D'Ettorre, 2010*). Analyses of CHC profiles showed that newly emerged honey bee workers express significantly lower amounts of total CHCs and lower overall CHC chemical diversity in comparison to older foragers, which are expected to elicit the strongest nestmate recognition response from guards at the entrance to the hive (*Breed et al., 2004*; *Kather et al., 2011*). Additionally, other studies have suggested that honey bee nestmate recognition cues might be derived from various environmental sources (*Downs and Ratnieks, 1999*), and hive building materials such as the honeycomb wax (*Breed, 1998*; *Breed et al., 1988*; *Couvillon et al., 2007*; *D'ettorre et al., 2006*). Based on these studies, it has been hypothesized that, similar to some ant species, the CHC profile of newly eclosed workers represents a 'blank slate' (*Breed et al., 2004*; *Lenoir et al., 1999*), and that nestmate recognition cues are subsequently acquired by individual workers primarily through the homogenization and transfer of chemicals via direct social interactions and intermediate environmental factors (*Breed et al., 2015*). Furthermore, it has recently been proposed that the cephalic salivary gland of honey bee workers is functionally analogous to the PPG in ants, and could be involved in the homogenization and transfer of the CHCs between colony members (*Martin et al., 2018*). However, when and how honey bee chemical nestmate recognition cues mature, and whether CHC homogenization mechanisms play a role in this process have not been directly investigated.

Here, we provide empirical evidence that the maturation of the CHC profile of individual honey bee workers is primarily regulated by innate developmental processes associated with age-dependent behavioral tasks and modulated by the social colony environment, and that mature colony-specific recognition cues are primarily associated with the foraging task. Specifically, we find that individual workers exhibit stereotypic quantitative and qualitative changes in their CHC profile as they transition from in-hive tasks to foraging outside, that these changes are associated with innate transcriptional changes in CHC biosynthetic pathway genes, and that only forager honey bees are behaviorally rejected from the entrance of an unrelated hive. Together, our findings suggest that not all members of honey bee colonies display a uniform cuticular chemical profile via the direct acquisition of CHC mixes. Instead, our data indicate that CHC profiles, and likely nestmate recognition cues, in honey bees are more likely a product of a genetically-determined developmental program that is modulated by colony-specific factors.

## Results

### CHC profiles of individual honey bee workers exhibit qualitative and quantitative age-dependent changes

Given that newly emerged honey bees have lower amounts of total CHCs, and exhibit less chemical diversity compared to older bees (*Breed et al., 2004*), we initially sought to determine the age at

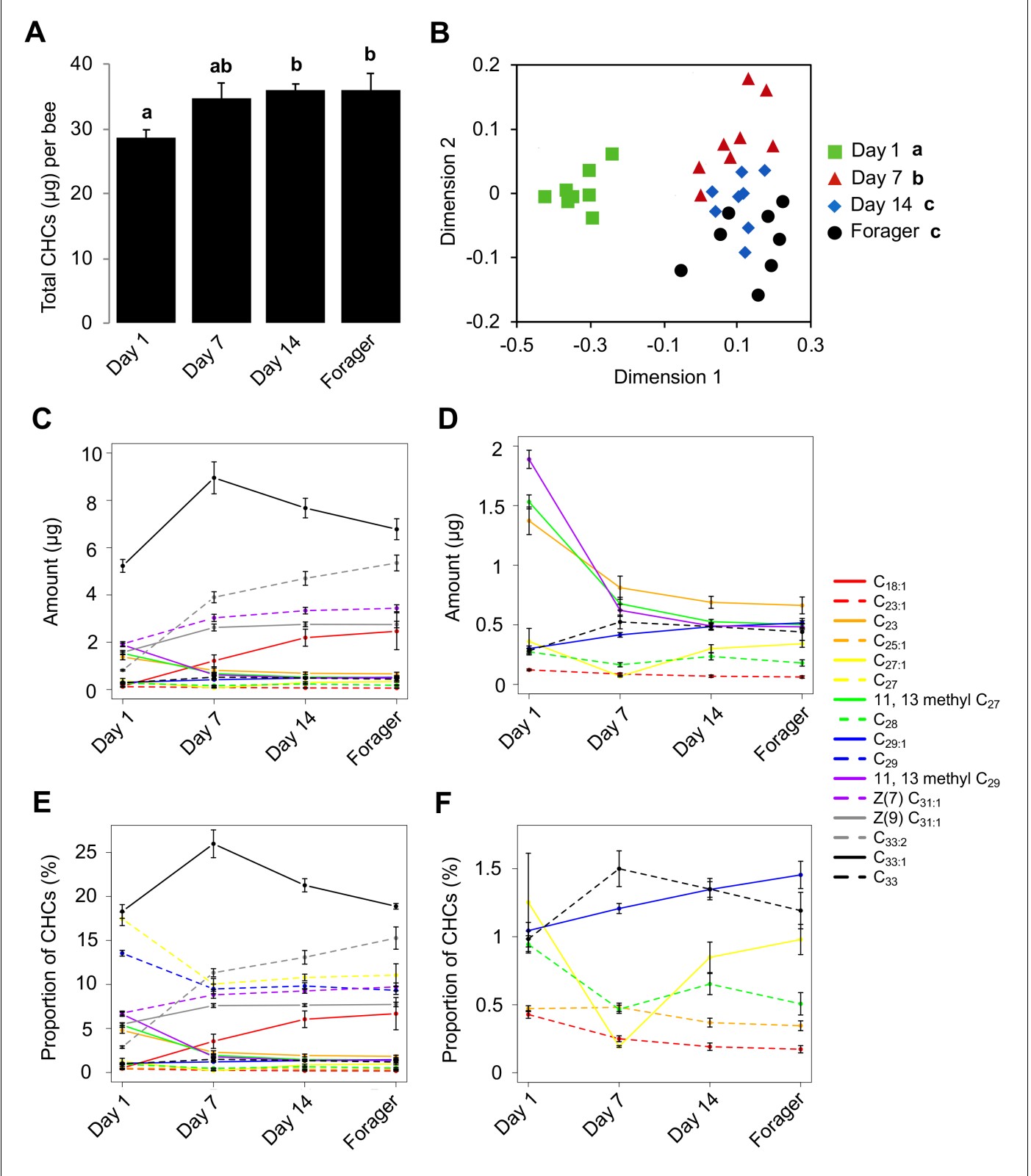

**Figure 1.** CHC profiles of bees exhibit quantitative and qualitative changes in association with age. (**A**) Total CHC amounts (µg) extracted from sister bees of different ages. (**B**) CHC profiles of sister bees of different ages. (**C**) Statistically significantly changing amounts (µg) of individual CHCs across sister bees of different ages. (**D**) A subset of C with low amounts. (**E**) Statistically significantly changing proportions of individual CHCs across sister bees of different ages. (**F**) A subset of C with low proportions. Statistics in A using ANOVA followed by Tukey's HSD post-hoc. Statistics in B using

*Figure 1 continued on next page*

*Figure 1 continued*

Permutation MANOVA followed by FDR pairwise contrasts shown as a non-metric multidimensional scaling plot depicting Bray-Curtis dissimilarity between samples. Statistics for C and D are listed in *Table 1*, statistics for E and F are listed in *Table 3*. Lowercase letters above bars in A and legend in B denote *posthoc* significance (p<0.05). Sample size per group, N = 8.

DOI: https://doi.org/10.7554/eLife.41855.003

The following source data and figure supplement are available for figure 1:

**Source data 1.** Amount (ng) of each compound extracted from each bee sample in *Figure 1* and *Figure 1—figure supplement 1*.
DOI: https://doi.org/10.7554/eLife.41855.005
**Figure supplement 1.** CHC profiles of bees exhibit quantitative and qualitative changes in association with age.
DOI: https://doi.org/10.7554/eLife.41855.004

which the CHC profile of individual honey bee workers matures. To achieve this goal, we analyzed the CHC profiles of individual workers from a single age-cohort that was reintroduced back into its source colony and then collected at different ages. This analysis revealed that the total amount of CHCs increases between one-day post-reintroduction and 14 days post-reintroduction and then

**Table 1.** Individual CHCs vary in total amount (ng) across different aged sister bees of a single colony.

Numbers represent mean amount (ng) of compound across bees of that age ±standard error. All p-values are from parametric ANOVA or nonparametric Kruskal Wallis ANOVA (denoted by 'KW'). Letters denote statistically significant age groups across individual compounds via Tukey's HSD (ANOVA post-hoc) or Dunn's Test with FDR adjustment (KW post-hoc) (p<0.05).

| Compound | Name | Retention time | Day 1 | Day 7 | Day 14 | Foraging | p-value |
|---|---|---|---|---|---|---|---|
| $C_{18:1}$ | z-(9)-Octadecenoic acid | 20.23 | 146.91 ± 52.98 (A) | 1213.9 ± 250.45 (AB) | 2193.35 ± 360.14 (B) | 2465.05 ± 779.73 (B) | 0.004 (KW) |
| $C_{23:1}$ | Tricosene | 20.72 | 121.69 ± 7.05 (A) | 86.74 ± 9.13 (B) | 68.64 ± 9.57 (B) | 61.79 ± 9.87 (B) | <0.001 |
| $C_{23}$ | Tricosane | 21.21 | 1372.77 ± 115.62 (A) | 811.16 ± 97.45 (B) | 688.47 ± 50.32 (B) | 662.47 ± 70.72 (B) | 0.001 (KW) |
| $C_{24}$ | Tetracosane | 22.79 | 43.54 ± 2.33 | 41.8 ± 3.58 | 46.47 ± 2.67 | 45.66 ± 4.39 | 0.756 |
| $C_{25:1}$ | Pentacosene | 23.89 | 134.75 ± 7.73 | 167.38 ± 15.77 | 131.94 ± 10.87 | 122.74 ± 13.46 | 0.123 (KW) |
| $C_{25}$ | Pentacosane | 24.34 | 1280.62 ± 78.61 | 1253.94 ± 145.03 | 1364.18 ± 51.8 | 1465.39 ± 195.17 | 0.663 |
| $C_{27:1}$ | Heptacosene | 26.87 | 359.94 ± 109.71 (A) | 67.55 ± 5.22 (B) | 299.98 ± 33.28 (A) | 340.49 ± 29.27 (A) | 0.006 |
| $C_{27}$ | Heptacosane | 27.3 | 5021.35 ± 335.43 | 3561.92 ± 394.09 | 3875.36 ± 135.84 | 4098.48 ± 688.78 | 0.071 (KW) |
| 11,13 methyl $C_{27}$ | 11- + 13 methyl Heptacosane | 27.75 | 1531.35 ± 58.53 (A) | 676.77 ± 53.97 (B) | 525.89 ± 19.46 (BC) | 501.91 ± 32.38 (C) | <0.001 |
| $C_{28}$ | Octacosane | 28.67 | 273.28 ± 26.17 (A) | 164.61 ± 17.67 (B) | 234.58 ± 29.25 (AB) | 179.57 ± 26.62 (AB) | 0.018 |
| $C_{29:1}$ | Nonacosene | 29.71 | 300.3 ± 21.91 (A) | 415.71 ± 19.53 (B) | 484.12 ± 21.65 (B) | 515.79 ± 37.33 (B) | <0.001 |
| $C_{29}$ | Nonacosane | 30.05 | 3891.37 ± 203.93 | 3361.26 ± 373.3 | 3535.4 ± 161.9 | 3421.62 ± 376.05 | 0.626 (KW) |
| 11,13 methyl $C_{29}$ | 11- + 13 methyl Nonacosane | 30.48 | 1888.38 ± 76.35 (A) | 621.6 ± 51.82 (B) | 490.47 ± 27.34 (B) | 482.86 ± 24.65 (B) | <0.001 |
| z(7) $C_{31:1}$ | z-(7)-Hentriacontene | 32.72 | 1930.38 ± 95.12 (A) | 3030.3 ± 152.07 (B) | 3336.99 ± 139.27 (B) | 3437.57 ± 146.12 (B) | <0.001 |
| z(9) $C_{31:1}$ | z-(9)Hentriacontene | 32.85 | 1572.04 ± 82.09 (A) | 2623.84 ± 145.02 (B) | 2756.09 ± 101.15 (B) | 2748.6 ± 137.39 (B) | <0.001 |
| $C_{31}$ | Hentriacontane | 33.25 | 2471.77 ± 155.09 | 3307.06 ± 347.6 | 3095.7 ± 150.22 | 2904.22 ± 351.69 | 0.158 (KW) |
| $C_{33:2}$ | Tritriacontadiene | 36.44 | 818.13 ± 30.77 (A) | 3901.71 ± 233.3 (B) | 4700.8 ± 289.64 (BC) | 5351.18 ± 332.47 (C) | <0.001 (KW) |
| $C_{33:1}$ | Tritriacotene | 37.13 | 5228.89 ± 272.19 (A) | 8952.18 ± 674.83 (B) | 7673.85 ± 413.67 (B) | 6778.42 ± 442.85 (AB) | <0.001 (KW) |
| $C_{33}$ | Tritriacotane | 37.76 | 282.16 ± 19.8 (A) | 523.87 ± 57.72 (B) | 485.29 ± 30.47 (B) | 440.35 ± 70.92 (AB) | 0.009 |

DOI: https://doi.org/10.7554/eLife.41855.006

**Table 2.** Individual CHCs vary in total amount (ng) across different aged sister bees of a second colony.

Numbers represent mean amount (ng) of compound across bees of that age ±standard error. All p-values are from parametric ANOVA or nonparametric Kruskal Wallis ANOVA (denoted by 'KW'). Letters denote statistically significant age groups across individual compounds via Tukey's HSD (ANOVA post-hoc) or Dunn's Test with FDR adjustment (KW post-hoc) (p<0.05).

| Compound | Name | Retention time | Day 1 | Day 7 | Day 14 | Foraging | p-value |
|---|---|---|---|---|---|---|---|
| $C_{18:1}$ | z-(9)-Octadecenoic acid | 20.23 | 969.4 ± 284.09 (A) | 4062.76 ± 288.4 (B) | 4410.73 ± 736.18 (B) | 5711.9 ± 1741.17 (B) | 0.004 (KW) |
| $C_{23:1}$ | Tricosene | 20.72 | 188.84 ± 31.5 (A) | 252.45 ± 41.06 (AB) | 449.27 ± 129.73 (AB) | 692.43 ± 192.5 (B) | 0.022 (KW) |
| $C_{23}$ | Tricosane | 21.21 | 2243.49 ± 329.45 | 1154.47 ± 276.64 | 3173.75 ± 1310.78 | 6778.99 ± 3385.43 | 0.108 (KW) |
| $C_{24}$ | Tetracosane | 22.79 | 92.03 ± 12.8 | 106.76 ± 34.59 | 191.62 ± 59.85 | 259.68 ± 81 | 0.078 (KW) |
| $C_{25:1}$ | Pentacosene | 23.89 | 223.72 ± 29.37 (A) | 369.19 ± 74.13 (AB) | 904.72 ± 314.93 (BC) | 1000.5 ± 254.7 (C) | 0.005 (KW) |
| $C_{25}$ | Pentacosane | 24.34 | 2897.27 ± 364.49 | 2820.94 ± 730.86 | 8160.17 ± 3143.65 | 8074.9 ± 3225.52 | 0.274 (KW) |
| $C_{27:1}$ | Heptacosene | 26.87 | 189.37 ± 62.24 (A) | 412.19 ± 85.56 (AB) | 804.57 ± 288.89 (B) | 635.81 ± 116.71 (B) | 0.017 (KW) |
| $C_{27}$ | Heptacosane | 27.3 | 10263.39 ± 1505.49 | 9644.33 ± 1394.55 | 15012.59 ± 3058.18 | 9042.02 ± 993.34 | 0.618 (KW) |
| 11,13 methyl $C_{27}$ | 11- + 13 methyl Heptacosane | 27.75 | 1373.16 ± 158.58 (A) | 722.54 ± 92.7 (AB) | 501.89 ± 82.05 (B) | 575.83 ± 217.27 (B) | 0.004 (KW) |
| $C_{28}$ | Octacosane | 28.67 | 284.5 ± 54.42 | 361.39 ± 56.38 | 430.28 ± 67.21 | 331.85 ± 35.45 | 0.348 |
| $C_{29:1}$ | Nonacosene | 29.71 | 420.31 ± 81.15 (A) | 1047.99 ± 126.61 (B) | 1491.92 ± 256 (B) | 925.21 ± 121.1 (B) | 0.002 (KW) |
| $C_{29}$ | Nonacosane | 30.05 | 5391.23 ± 1071 | 10175 ± 1632.19 | 12313.81 ± 2374.12 | 8962.64 ± 1385.72 | 0.087 |
| 11,13 methyl $C_{29}$ | 11- + 13 methyl Nonacosane | 30.48 | 1702.45 ± 166.85 (A) | 800.59 ± 95.67 (A) | 535.16 ± 66.24 (AB) | 382.5 ± 50.73 (B) | <0.001 |
| z(7) $C_{31:1}$ | z-(7)-Hentriacontene | 32.72 | 2189.03 ± 291.48 (A) | 5370.16 ± 756.21 (B) | 6185.18 ± 761.39 (B) | 4426.13 ± 737.85 (AB) | 0.005 |
| z(9) $C_{31:1}$ | z-(9)Hentriacontene | 32.85 | 1465.83 ± 254.41 (A) | 4120.89 ± 652.81 (B) | 4581.47 ± 665.27 (B) | 3361.18 ± 661.26 (AB) | 0.012 |
| $C_{31}$ | Hentriacontane | 33.25 | 3796.75 ± 847.96 | 10066.5 ± 1716.97 | 10082.89 ± 2677.18 | 7342.33 ± 1569.25 | 0.121 |
| $C_{33:2}$ | Tritriacontadiene | 36.44 | 242.82 ± 53.78 (A) | 1724.73 ± 289.89 (B) | 1353.86 ± 175.8 (B) | 1393.43 ± 313.9 (B) | 0.002 |
| $C_{33:1}$ | Tritriacotene | 37.13 | 7454.8 ± 1388.34 (A) | 18126.35 ± 2516.57 (B) | 17031.3 ± 3257.55 (AB) | 9630.94 ± 1791.06 (AB) | 0.014 |
| $C_{33}$ | Tritriacotane | 37.76 | 540.28 ± 149 (A) | 2165.09 ± 457.66 (B) | 1734.79 ± 523.36 (AB) | 1214.4 ± 271.66 (AB) | 0.037 (KW) |

DOI: https://doi.org/10.7554/eLife.41855.007

remains stable (*Figure 1A*, Kruskal-Wallis, H = 9.21, df = 3, p=0.026, FDR pairwise contrasts: Day 1 vs. Day 7 p=0.11, Day 1 vs. Day 14 p=0.036, Day 1 vs. Day 21 p=0.04, Day 7 vs. Day 14 = 0.613, Day 7 vs. Day 21 p=0.691, Day 14 vs. Day 21 p=0.79; *Figure 1—figure supplement 1A*, ANOVA, F (3,28) = 6.40, p=0.002, FDR pairwise contrasts: Day 1 vs. Day 7 p=0.036, Day 1 vs. Day 14 p=0.001, Day 1 vs. Day 18 p=0.007, Day 7 vs. Day 14 = 0.412, Day 7 vs. Day 18 p=0.993, Day 14 vs. Day 18 p=0.305). Additionally, individual compounds vary in total amount across bees of different ages (*Figure 1C,D*, *Figure 1—figure supplement 1C,D*, *Table 1*, *Table 2*). Independently of the age-related quantitative changes, we also found that the CHC profiles of workers exhibit age-related qualitative changes in the overall CHC chemical composition (*Figure 1B*, Permutation MANOVA, F (1,31) = 22.86, $R^2$ = 0.43, p<0.001, FDR pairwise contrasts: Day 1 vs. Day 7 p=0.002, Day 1 vs. Day 14 p=0.002, Day 1 vs. Day 21 p=0.002, Day 7 vs. Day 14 = 0.017, Day 7 vs. Day 21 p=0.002, Day 14 vs. Day 21 p=0.31; *Figure 1—figure supplement 1B*, Permutation MANOVA, F(3,28) = 2.35,

**Table 3.** Individual CHCs vary in proportion across different aged sister bees of a single colony.

Numbers represent mean percentage of compound across bees of that age ±standard error. All p-values are from parametric ANOVA or nonparametric Kruskal Wallis ANOVA (denoted by 'KW'). Letters denote statistically significant age groups across individual compounds via Tukey's HSD (ANOVA post-hoc) or Dunn's Test with FDR adjustment (KW post-hoc) (p<0.05).

| Compound | Name | Retention time | Day 1 | Day 7 | Day 14 | Foraging | p-value |
|---|---|---|---|---|---|---|---|
| $C_{18:1}$ | z-(9)-Octadecenoic acid | 20.23 | 0.51 ± 0.19 (A) | 3.55 ± 0.81 (B) | 6.05 ± 0.95 (B) | 6.68 ± 1.83 (B) | <0.001 (KW) |
| $C_{23:1}$ | Tricosene | 20.72 | 0.43 ± 0.03 (A) | 0.25 ± 0.02 (B) | 0.19 ± 0.03 (B) | 0.17 ± 0.03 (B) | <0.001 (KW) |
| $C_{23}$ | Tricosane | 21.21 | 4.79 ± 0.33 (A) | 2.30 ± 0.18 (B) | 1.93 ± 0.16 (B) | 1.83 ± 0.14 (B) | <0.001 |
| $C_{24}$ | Tetracosane | 22.79 | 0.15 ± 0.01 | 0.12 ± 0.01 | 0.13 ± 0.01 | 0.13 ± 0.01 | 0.051 |
| $C_{25:1}$ | Pentacosene | 23.89 | 0.47 ± 0.02 (A) | 0.48 ± 0.03 (A) | 0.37 ± 0.03 (AB) | 0.35 ± 0.04 (B) | 0.006 |
| $C_{25}$ | Pentacosane | 24.34 | 4.46 ± 0.17 | 3.54 ± 0.23 | 3.82 ± 0.19 | 4.02 ± 0.35 | 0.051 (KW) |
| $C_{27:1}$ | Heptacosene | 26.87 | 1.25 ± 0.36 (A) | 0.19 ± 0.01 (B) | 0.85 ± 0.11 (A) | 0.98 ± 0.11 (A) | 0.002 (KW) |
| $C_{27}$ | Heptacosane | 27.30 | 17.45 ± 0.77 (A) | 10.05 ± 0.62 (B) | 10.79 ± 0.38 (B) | 11.06 ± 1.29 (B) | 0.001 (KW) |
| 11,13 methyl $C_{27}$ | 11- + 13 methyl Heptacosane | 27.75 | 5.38 ± 0.24 (A) | 1.95 ± 0.09 (B) | 1.47 ± 0.07 (B) | 1.43 ± 0.11 (B) | <0.001 |
| $C_{28}$ | Octacosane | 28.67 | 0.94 ± 0.06 (A) | 0.47 ± 0.03 (B) | 0.65 ± 0.08 (AB) | 0.51 ± 0.08 (B) | 0.001 (KW) |
| $C_{29:1}$ | Nonacosene | 29.71 | 1.05 ± 0.06 (A) | 1.21 ± 0.04 (AB) | 1.35 ± 0.06 (B) | 1.46 ± 0.10 (B) | 0.001 |
| $C_{29}$ | Nonacosane | 30.05 | 13.58 ± 0.33 (A) | 9.49 ± 0.55 (B) | 9.82 ± 0.37 (B) | 9.34 ± 0.47 (B) | <0.001 |
| 11,13 methyl $C_{29}$ | 11- + 13 methyl Nonacosane | 30.48 | 6.62 ± 0.24 (A) | 1.79 ± 0.08 (B) | 1.37 ± 0.08 (B) | 1.37 ± 0.08 (B) | <0.001 |
| z(7) $C_{31:1}$ | z-(7)-Hentriacontene | 32.72 | 6.75 ± 0.23 (A) | 8.82 ± 0.39 (B) | 9.26 ± 0.26 (B) | 9.72 ± 0.44 (B) | <0.001 |
| z(9) $C_{31:1}$ | z-(9)Hentriacontene | 32.85 | 5.48 ± 0.16 (A) | 7.60 ± 0.23 (B) | 7.65 ± 0.16 (B) | 7.73 ± 0.26 (B) | <0.001 |
| $C_{31}$ | Hentriacontane | 33.25 | 8.58 ± 0.26 | 9.39 ± 0.55 | 8.61 ± 0.37 | 7.90 ± 0.51 | 0.148 |
| $C_{33:2}$ | Tritriacontadiene | 36.44 | 2.88 ±. 14 (A) | 11.33 ± 0.46 (B) | 13.08 ± 0.77 (B) | 15.27 ± 1.26 (B) | <0.001 (KW) |
| $C_{33:1}$ | Tritriacotene | 37.13 | 18.29 ± 0.78 (A) | 25.97 ± 1.58 (B) | 21.26 ± 0.75 (A) | 18.88 ± 0.32 (A) | <0.001 |
| $C_{33}$ | Tritriacotane | 37.76 | 0.98 ± 0.06 (A) | 1.50 ± 0.13 (B) | 1.35 ± 0.08 (AB) | 1.19 ± 0.13 (AB) | 0.012 |

DOI: https://doi.org/10.7554/eLife.41855.008

$R^2$ = 0.22, p=0.038, FDR pairwise contrasts: Day 1 vs. Day 7 p=0.024, Day 1 vs. Day 14 p=0.011, Day 1 vs. Day 18 p=0.018, Day 7 vs. Day 14 = 0.406, Day 7 vs. Day 18 p=0.212, Day 14 vs. Day 18 p=0.524), as well as in the relative amounts of individual CHCs (*Figure 1E,F*, *Figure 1—figure supplement 1E,F*, *Table 3*, *Table 4*). These data confirm that not all members of a honey bee colony share a common CHC profile (*Kather et al., 2011*), and suggest that age-dependent processes might be playing an important role in the regulation of both the quantitative and qualitative dimensions of the cuticular chemical profiles of individual honey bee workers.

## The CHC profiles of individual workers are task-related

Honey bee workers exhibit age-related division of labor, which is characterized by a stereotypic sequence of in-hive behavioral tasks such as nursing and food handling, followed by the final transition to foraging outside the colony at about three weeks of age (*Robinson, 1992*; *Smith et al., 2008*; *Søvik et al., 2015*). Consequently, under natural colony settings, it is impossible to separate the possible independent impacts of 'age' and 'task' on the expression of forager-specific CHC profiles. Therefore, we next analyzed the CHC profiles of individual nurse and forager bees from single-cohort-colonies (SCC), a well-established experimental approach to uncouple behavioral maturation from chronological age (*Ben-Shahar et al., 2004*; *Ben-Shahar et al., 2002*; *Greenberg et al., 2012*; *Robinson et al., 1989*; *Whitfield et al., 2003*). Because these artificial colonies are initially comprised of a single age-cohort of day-old bees, a small proportion of these young workers will accelerate their behavioral maturation to become precocious foragers that are the same age as typical nurses (~7 days old) (*Ben-Shahar et al., 2002*; *Greenberg et al., 2012*; *Huang and Robinson, 1992*). The comparison of the CHC profiles of typical young nurses and precocious foragers of identical age revealed a significant effect of task on the CHC profile of individual workers (*Figure 2A*, Permutation MANOVA, F(1,15) = 13.79, $R^2$ = 0.50, p<0.001). Similarly, we observed a significant

**Table 4.** Individual CHCs vary in proportion across different aged sister bees of a second colony.

Numbers represent mean percentage of compound across bees of that age ±standard error. All p-values are from parametric ANOVA or nonparametric Kruskal Wallis ANOVA (denoted by 'KW'). Letters denote statistically significant age groups across individual compounds via Tukey's HSD (ANOVA post-hoc) or Dunn's Test with FDR adjustment (KW post-hoc) (p<0.05).

| Compound | Name | Retention time | Day 1 | Day 7 | Day 14 | Foraging | p-value |
|---|---|---|---|---|---|---|---|
| $C_{18:1}$ | z-(9)-Octadecenoic acid | 20.23 | 2.47 ± 0.64 (A) | 6.06 ± 0.91 (B) | 5.34 ± 0.98 (AB) | 7.66 ± 1.78 (B) | 0.020 (KW) |
| $C_{23:1}$ | Tricosene | 20.72 | 0.47 ± 0.06 (AB) | 0.36 ± 0.07 (A) | 0.51 ± 0.13 (AB) | 0.95 ± 0.23 (B) | 0.034 |
| $C_{23}$ | Tricosane | 21.21 | 5.61 ± 0.71 (A) | 1.82 ± 0.63 (B) | 3.59 ± 1.50 (AB) | 9.13 ± 4.29 (AB) | 0.017 (KW) |
| $C_{24}$ | Tetracosane | 22.79 | 0.13 ± 0.03 | 0.11 ± 0.02 | 0.09 ± 0.02 | 0.12 ± 0.02 | 0.662 |
| $C_{25:1}$ | Pentacosene | 23.89 | 0.54 ± 0.04 | 0.55 ± 0.14 | 1.02 ± 0.32 | 1.40 ± 0.32 | 0.050 (KW) |
| $C_{25}$ | Pentacosane | 24.34 | 7.09 ± 0.61 | 4.55 ± 1.73 | 9.17 ± 3.54 | 11.10 ± 4.18 | 0.106 (KW) |
| $C_{27:1}$ | Heptacosene | 26.87 | 0.28 ± 0.065 (A) | 0.62 ± 0.17 (AB) | 0.92 ± 0.29 (AB) | 0.91 ± 0.15 (B) | 0.030 (KW) |
| $C_{27}$ | Heptacosane | 27.3 | 24.41 ± 0.59 (A) | 13.81 ± 2.39 (B) | 16.22 ± 2.35 (B) | 12.97 ± 1.28 (B) | 0.012 (KW) |
| 11,13 methyl $C_{27}$ | 11- + 13 methyl Heptacosane | 27.75 | 3.76 ± 0.83 (A) | 0.93 ± 0.07 (A) | 0.57 ± 0.08 (B) | 0.44 ± 0.08 (B) | <0.001 (KW) |
| $C_{28}$ | Octacosane | 28.67 | 0.68 ± 0.11 | 0.48 ± 0.04 | 0.48 ± 0.05 | 0.49 ± 0.06 | 0.347 (KW) |
| $C_{29:1}$ | Nonacosene | 29.71 | 0.98 ± 0.08 | 0.93 ± 0.21 | 1.71 ± 0.24 | 1.23 ± 0.32 | 0.095 |
| $C_{29}$ | Nonacosane | 30.05 | 12.39 ± 0.80 | 13.18 ± 1.15 | 13.80 ± 2.08 | 13.14 ± 2.02 | 0.952 |
| 11,13 methyl $C_{29}$ | 11- + 13 methyl Nonacosane | 30.48 | 4.62 ± 0.95 (A) | 1.11 ± 0.09 (A) | 0.60 ± 0.05 (B) | 0.58 ± 0.10 (B) | <0.001 (KW) |
| z(7) $C_{31:1}$ | z-(7)-Hentriacontene | 32.72 | 5.27 ± 0.30 | 7.30 ± 0.75 | 7.20 ± 0.96 | 6.65 ± 1.29 | 0.450 |
| z(9) $C_{31:1}$ | z-(9)Hentriacontene | 32.85 | 3.43 ± 0.24 | 5.47 ± 0.62 | 5.25 ± 0.71 | 4.99 ± 1.04 | 0.257 |
| $C_{31}$ | Hentriacontane | 33.25 | 8.64 ± 0.72 | 13.02 ± 1.52 | 10.97 ± 2.13 | 10.50 ± 2.03 | 0.418 |
| $C_{33:2}$ | Tritriacontadiene | 36.44 | 0.55 ± 0.07 (A) | 2.38 ± 0.35 (B) | 1.58 ± 0.22 (AB) | 2.09 ± 0.49 (B) | 0.006 |
| $C_{33:1}$ | Tritriacotene | 37.13 | 17.47 ± 1.31 | 24.51 ± 2.83 | 19.14 ± 3.06 | 13.95 ± 2.38 | 0.056 |
| $C_{33}$ | Tritriacotane | 37.76 | 1.21 ± 0.16 | 2.83 ± 0.51 | 1.84 ± 0.36 | 1.73 ± 0.35 | 0.116 (KW) |

DOI: https://doi.org/10.7554/eLife.41855.009

effect of task on the CHC profiles of individual 'over-aged' nurses and typical-aged foragers at three weeks of age (*Figure 2B*, Permutation MANOVA, F(1,15) = 45.41, $R^2$ = 0.76, p<0.001). In contrast, task and age had no effect on total CHC amount (*Figure 2C*, Two-way ANOVA, age: F(1,28) = 0.55, p=0.46, task: F(1,28) = 0.37, p=0.55, age*task: F(1,28) = 5.37, p=0.03). Together, these data suggest that processes associated with the behavioral maturation of honey bee workers, not chronological age, are primarily responsible for the observed forager versus nurse CHC profiles of individual honey bee workers.

Previous studies in Harvester ants suggested that exposure to the environment outside the nest is sufficient to induce stereotypical changes in the CHC profiles of individual social insects (*Wagner et al., 2001*). Therefore, we next asked whether spending time outside the hive is sufficient to induce the observed forager-specific CHC profile by comparing the CHC profiles between 'undertakers', nurses, and foragers from typical colonies. 'Undertakers' are a small group of highly specialized older pre-foraging workers (2–3 weeks of age), which are responsible for removing dead bees by carrying them outside and away from the colony (*Robinson, 1992*; *Smith et al., 2008*; *Søvik et al., 2015*; *Trumbo et al., 1997*). Therefore, because undertakers and foragers perform their respective tasks outside the hive, while nurses and other younger, pre-foraging bees rarely do, we reasoned that if outdoor exposure defines the distinct forager-specific CHC profile then the CHC profiles of undertakers should be more similar to foragers than to nurses. However, we found that the CHC profiles of undertakers are markedly different from those of foragers, and are more similar to those of nurses (*Figure 2D*, Permutation MANOVA, F(2,23)=12.60, $R^2$ = 0.55, p<0.001, FDR pairwise contrasts: undertaker vs. forager p=0.003, undertaker vs. nurse p=0.176, forager vs. nurse p=0.003). These data suggest that some outdoor exposure is not sufficient to drive forager-specific CHC profiles.

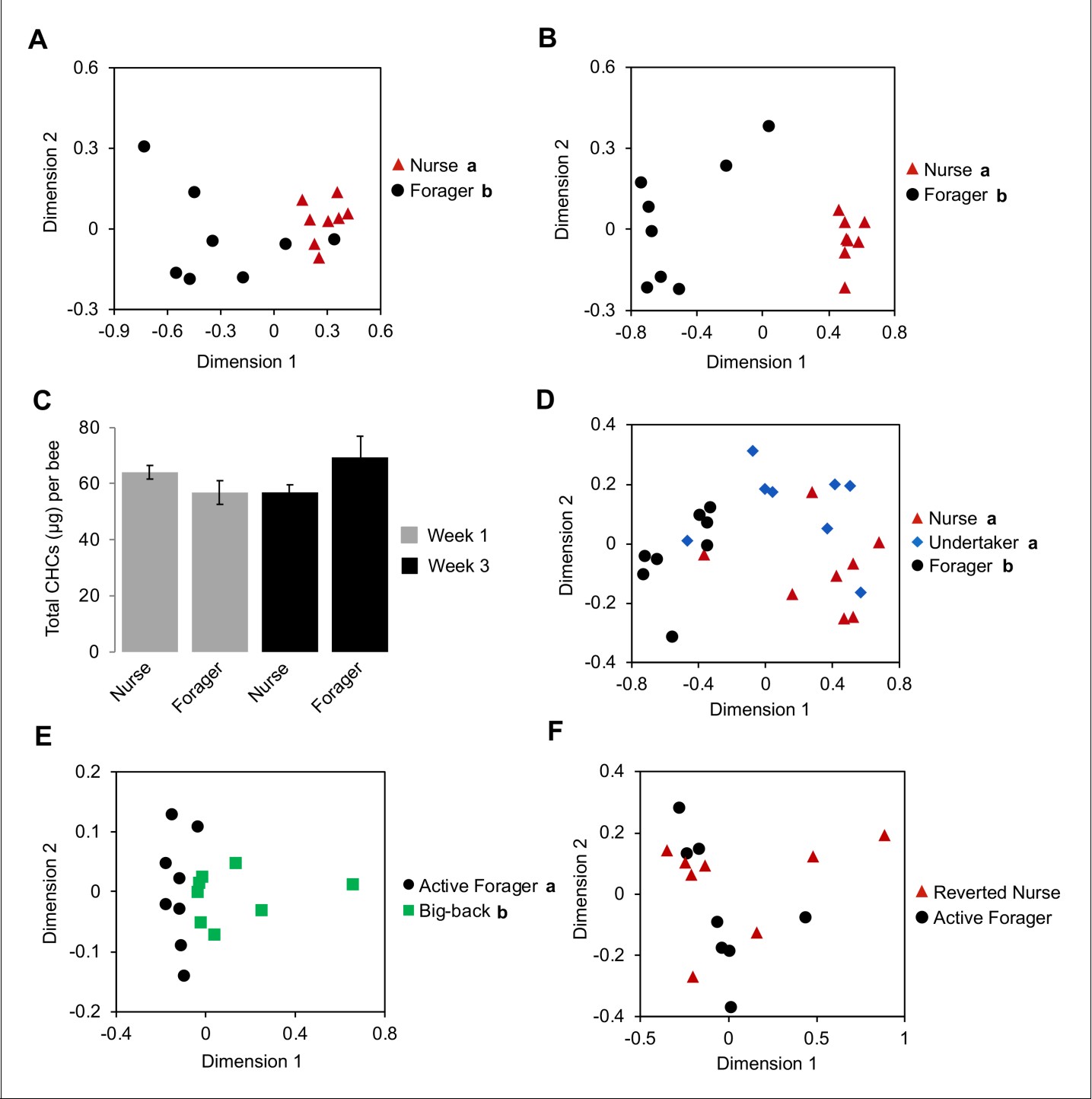

**Figure 2.** Effect of task on the CHC profile of bees is independent of age. Single cohort colony bees differ in CHC profile by behavioral task at one week of age (typical nurse age, (**A**)) and three weeks of age (typical forager age, (**B**)). (**C**) SCC bees do not differ in total CHC amount due to age and/or task. (**D**) Undertakers and nurses differ from foragers in CHC profile. (**E**) 'Big-back' bees differ from same-aged actively foraging sisters in CHC profile. Total CHC statistics (**C**) using ANOVA followed by Tukey's HSD with FDR correction. CHC profile statistics (**A, B, D, E**) using Permutation MANOVA followed by FDR pairwise contrasts, shown as non-metric multidimensional scaling plots depicting Bray-Curtis dissimilarity between samples. Letters in graphs and legends denote *posthoc* statistical significance ($p<0.05$). Sample size per group, N = 8.
DOI: https://doi.org/10.7554/eLife.41855.010

The following source data and figure supplement are available for figure 2:

**Source data 1.** Amount (ng) of each compound extracted from each bee sample in *Figure 2* and *Figure 2—figure supplement 1*.

*Figure 2 continued*
DOI: https://doi.org/10.7554/eLife.41855.012
**Figure supplement 1.** CHC profiles differ between unknown-aged foragers from two different colonies at a single location.
DOI: https://doi.org/10.7554/eLife.41855.011

We next asked whether the CHC profiles of foragers are a direct consequence of their behavioral state by using 'big back colonies' (*Ben-Shahar et al., 2000*; *Withers et al., 1995*), which allowed us to compare active foragers to bees of a similar age and behavioral state that are unable to forage outside (see Materials and methods). We found that the overall CHC profiles of 'big-back' bees were different from those of their actively foraging sisters (*Figure 2E*, Permutation MANOVA, $F(1,14)=5.91$, $R^2 = 0.313$, $p<0.001$). These data suggest that the physiological transition to foraging behaviors is not the sole factor that defines forager-specific CHC profiles, and that it could be modulated by additional factors associated with the act of foraging itself and/or extended exposure to various outdoor environmental factors. However, the fact that foraging nestmates express very similar CHC profiles, which are markedly different from those of non-nestmate foragers sharing a similar foraging environment (*Figure 2—figure supplement 1*, Permutation MANOVA, $F(1,15) = 12.5$, $R^2 = 0.47$, $p<0.001$) suggests that forager-specific CHC profiles are not simply defined by the foraging environment. Additionally, to test whether extended exposure to outdoor environmental factors induces predictable changes in CHCs, we compared the relative amounts of individual compounds

**Table 5.** CHCs vary in relative proportion between foragers and in-hive bees across studies.
Numbers represent difference in mean percentage of each compound in forager bees relative to in-hive bees. 'Hive 1' denotes forager bees minus Day 14 bees corresponding to *Figure 1*; 'Hive 2' denotes forager bees minus Day 14 bees corresponding to *Figure 1—figure supplement 1*; 'SCC week 1' and 'SCC week 3' denote forager bees minus nurse bees corresponding to *Figure 2A and B*, respectively; 'Undertaker' denotes forager bees minus undertaker bees corresponding to *Figure 2D*; 'Big-back' denotes forager bees minus big-back bees corresponding to *Figure 2E*. Statistics using Student's t-test or Mann-Whitney U between the forager and in-hive bee group. Asterisks (*) or plus sign (+) denote statistical significance for t-test or Mann-Whitney U, respectively. 'ns' denotes non-significant differences.

| Compound | Hive 1 | Hive 2 | SCC 1 week | SCC 3 weeks | Undertaker | 'Big-back' |
|---|---|---|---|---|---|---|
| $C_{18:1}$ | ns | ns | 3.71 (+) | ns | −8.86 (*) | ns |
| $C_{23:1}$ | ns | ns | 0.51 (+) | 1.56 (+) | 0.78 (+) | ns |
| $C_{23}$ | ns | ns | 3.48 (+) | 11.45 (*) | 10.8 (+) | −0.85 (+) |
| $C_{24}$ | ns | ns | 0.25 (+) | 0.54 (+) | 0.52 (*) | −0.09 (+) |
| $C_{25:1}$ | ns | ns | 1.28 (+) | 4.15 (+) | 1.66 (*) | ns |
| $C_{25}$ | ns | ns | 11.58 (+) | 24.22 (+) | 21.12 (+) | −3.83 (+) |
| $C_{27:1}$ | ns | ns | 1.43 (+) | 2.18 (+) | ns | −0.29 (+) |
| $C_{27}$ | ns | ns | 8.83 (*) | 9.38 (*) | ns | −8.35 (+) |
| 11,13 methyl $C_{27}$ | ns | ns | ns | −0.35 (*) | −0.23 (+) | −0.45 (*) |
| $C_{28}$ | ns | ns | ns | ns | −0.07 (*) | −0.05 (*) |
| $C_{29:1}$ | ns | ns | 0.69 (+) | ns | −1.08 (*) | ns |
| $C_{29}$ | ns | ns | −4.19 (*) | −3.31 (*) | −2.58 (*) | ns |
| 11,13 methyl $C_{29}$ | ns | ns | ns | −0.24 (+) | −0.52 (*) | ns |
| z(7) $C_{31:1}$ | ns | ns | −2.92 (*) | −5.53 (+) | −4.78 (+) | 1.6 (+) |
| z(9) $C_{31:1}$ | ns | ns | −2.94 (*) | −5.23 (+) | −2.93 (+) | ns |
| $C_{31}$ | ns | ns | −6.09 (+) | −6.75 (+) | −2.12 (*) | ns |
| $C_{33:2}$ | ns | ns | −1.25 (+) | −6.63 (+) | −2.9 (+) | ns |
| $C_{33:1}$ | −2.38 (*) | ns | −13.81 (*) | −21.39 (+) | −8.07 (+) | 6.65 (*) |
| $C_{33}$ | ns | ns | −1.1 (+) | −1.56 (+) | −0.48 (*) | |

DOI: https://doi.org/10.7554/eLife.41855.013

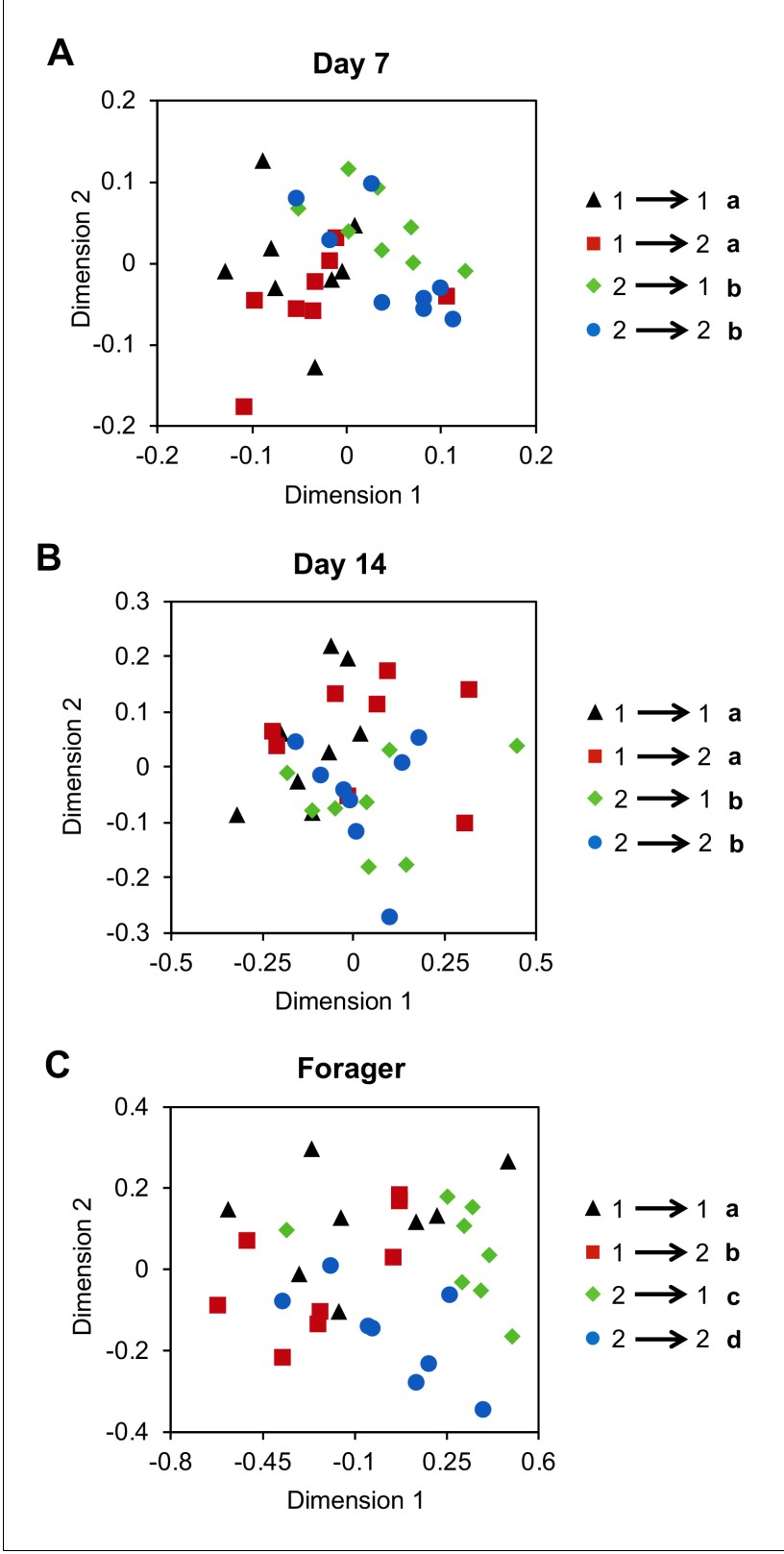

**Figure 3.** Cross-fostering indicates colony environment drives the signature CHC profiles of foragers. Age-matched cross-fostered bees differ in CHC profile by source colony at Day 7 (**A**) and Day 14 (**B**), and by both source colony and foster colony when they are foragers (**C**). Number to left of arrow in legend represents the bee's source colony, and the number to the right represents the bee's foster colony. All statistics using
*Figure 3 continued on next page*

*Figure 3 continued*

Permutation MANOVA followed by FDR pairwise contrasts, shown as non-metric multidimensional scaling plots depicting Bray-Curtis dissimilarity between samples. Letters in legends denote *posthoc* statistical significance (p<0.05). Sample size per group, N = 8.

DOI: https://doi.org/10.7554/eLife.41855.014

The following source data and figure supplement are available for figure 3:

**Source data 1.** Amount (ng) of each compound extracted from each bee sample in *Figure 3*.

DOI: https://doi.org/10.7554/eLife.41855.016

**Figure supplement 1.** Sample size assessment of cross-fostered bees indicates sample size of 8 is adequate.

DOI: https://doi.org/10.7554/eLife.41855.015

between forager bees and in-hive bees across our various experiments. We did not find a single compound that varied between foragers and in-hive bees in a consistent manner across our experiments (e.g. always increases or always decreases in association with foraging activity) (*Table 5*), indicating that CHCs do not change in a stereotypic manner in association with extended outdoor exposure, as they do in Harvester ants (*Wagner et al., 2001*). Nevertheless, to further examine whether forager-specific CHC profiles were solely environmentally determined, we also analyzed the CHC profiles of typical-age foragers that were forced to revert to a nursing state (*Robinson et al., 1992*). However, we did not find any differences between the CHC profiles of reverted nurses and active foragers (*Figure 2F*). These data suggest that once foragers acquire their signature CHC profile, it remains stable independent of the task they perform and despite the typical short CHC half-life in insects (*Kent et al., 2007*). Together, these data suggest that forager-specific CHC profiles are derived from a combination of factors associated with an innate behavioral maturation process, as well as being physically engaged in foraging activity.

## The development of individual CHC profiles is a regulated process modulated by the colony environment

Previous work indicates that guard bees will accept foraging-age nestmates and reject foraging-age non-nestmates, independent of genetic relatedness (*Downs and Ratnieks, 1999*). This suggests that factors associated with the hive environment play a dominant role in specifying the colony-specific chemical signatures used for nestmate recognition. Yet, our data also indicate that CHC profile development in individual workers is a developmentally-regulated process that is closely associated with the age-dependent division of labor among workers. To address this potential conundrum, we next asked whether the effects of task and colony environment on the development of CHC profiles of individual workers are independent by using a reciprocal cross-fostering strategy. To achieve our goal, we introduced cohorts of newly eclosed bees from two different typical colonies back into their source colony, as well as a reciprocal foster colony, and then recollected marked workers from both cohorts in each reciprocal colony at different ages. CHC analyses revealed that through Day 14, the CHC profiles of bees were more similar to the profiles of their same-aged non-nestmate sisters than those of unrelated nestmates of similar age (*Figure 3A*, Two-way Permutation MANOVA, foster colony (environment): $F_{(1,31)} = 2.19$, $R^2 = 0.06$, p=0.06, source colony (genetics): $F_{(1,31)} = 5.94$, $R^2 = 0.16$, p<0.001, foster colony*source colony: $F_{(1,31)}=0.46$, $R^2 = 0.01$, p=0.82; *Figure 3B*, Two-way Permutation MANOVA, foster colony: $F_{(1,31)} = 1.13$, $R^2 = 0.03$, p=0.33, source colony: $F_{(1,31)} = 3.18$, $R^2 = 0.09$, p=0.02, foster colony*source colony: $F_{(1,31)} = 1.78$, $R^2 = 0.05$, p=0.15; sample size assessment depicted in *Figure 3—figure supplement 1* indicates sample size is adequate). In contrast, once workers shift to foraging activity, we found that the CHC profiles of fostered bees are different from the profiles of both foraging sisters raised in the source colony and unrelated host foragers of similar age (*Figure 3C*, Two-way Permutation MANOVA, foster colony: $F_{(1,31)} = 4.04$, $R^2 = 0.10$, p=0.02, source colony: $F_{(1,31)} = 7.65$, $R^2 = 0.19$, p=0.001, foster colony*source colony: $F_{(1,31)}=0.48$, $R^2 = 0.01$, p=0.67). Together, these data suggest that genetic variations, or other long-term effects associated with the source colony, play an important role in defining the CHC profiles of individuals during the early phases of the age-dependent behavioral development of worker honey bees. However, by the time bees start foraging, the mature CHC profile of individual workers is defined by an interaction between factors associated with both the source and foster colonies (*Figure 3C*).

**Table 6.** CHC biosynthesis genes, BLAST E-values, and quantitative real-time PCR primers used in this study. BLAST E-values listed with known *Drosophila melanogaster* enzyme gene compared to.

| Gene | Function | BLAST E-value (*D. melanogaster* gene) | Forward primer | Reverse primer | Previously published |
|---|---|---|---|---|---|
| LOC724867 | Elongase | 2E-37 (*EloF*) | TGGGACCGGAATATCAAAAA | GCAGTAAAAGTGCCGCTACC | *Falcón et al. (2014)* |
| LOC724552 | Elongase | 1E-38 (*EloF*) | TCGGTAATCATGGAGTTATATAAGGA | ATCTTGGTCCAGCTGATAAGG | *Falcón et al. (2014)* |
| LOC550828 | Elongase | 1E-37 (*EloF*) | TCGTCAAAGTTTTGGGTCCT | GACCTCCCCATCCTGCTATC | |
| LOC409638 | Elongase | 9E-38 (*EloF*) | TGGATCGATTCCACGAGATA | CATCAGCTTTGGCCCTAAAA | |
| LOC413789 | Elongase | 2E-38 (*EloF*) | CAGATCTGGTGCACGGGTA | TTCTCCATTATCCTCGGTCCT | *Falcón et al. (2014)* |
| LOC100578829 | Elongase | 1E-36 (*EloF*) | ATGGCCTCGTTCGGTATTTT | ACGAATTGGACCATTTGCAC | *Falcón et al. (2014)* |
| LOC726467 | Elongase | 2E-13 (*Elo68α*) | GAGTTCATTACTTTCATTGTTTTCCA | AACATCCATGACCAAAAACCA | *Falcón et al. (2014)* |
| LOC725842 | Elongase | 5E-10 (*Elo68α*) | ATTAACGTATCACGGTTTTTATCAT | TTAATTCCTGCTTTCGTAACACT | *Falcón et al. (2014)* |
| LOC725031 | Elongase | 1E-8 (*Elo68α*) | TGGAACACATTGCTTGCATC | TGTCCAAAAACCAGACACGA | *Falcón et al. (2014)* |
| GB51249 | Elongase | 7E-7 (*EloF*) | ATGTCGATTTTAATGCAATACGTG | AAACTTTTACACCATATACGTAGCTCA | *Falcón et al. (2014)* |
| LOC412166 | Desaturase | 5E-174 (*desat1*) | CGCTGCTCATATCTTTGGAA | ATTTCCCAATTCTGCCGTTT | *Falcón et al. (2014)* |
| LOC551527 | Desaturase | 1E-138 (*desat1*) | TTAATGGTCCGAAAGCATCC | CCCATGTAGGAATTACAAAGCA | *Falcón et al. (2014)* |
| LOC552417 | Desaturase | 2E-137 (*desat1*) | TACGTTTCGTGCTGATGCTT | ACCAACCCATATGCGAGAAG | *Falcón et al. (2014)* |
| LOC100576797 | Desaturase | 8E-127 (*desat1*) | ACGGGTGAACTTGGTGGTTA | TTTTGTTGCAGCTCGATTCA | *Falcón et al. (2014)* |
| LOC552176 | Desaturase | 2E-113 (*desat1*) | ACTACCGGATTCGGCATAACT | CTGTGATCCAATGCCCATCT | *Falcón et al. (2014)* |
| LOC727166 | Desaturase | 4E-56 (*desat1*) | TGGTCTGGAATATCAAGGAAGG | ACCGAATTCACCACATTTCC | |
| LOC727333 | Desaturase | 4E-54 (*desat1*) | GGGCCCATAAAACATACGAA | TGTATGGATCTTTATCAGTCCCATAAT | *Falcón et al. (2014)* |
| eIF3-S8 | Eukaryotic translation initiation factor | | TCTTGGACCAGCAGTAGCAG | GCATATCGAGCATTTCCGTA | |

DOI: https://doi.org/10.7554/eLife.41855.019

## The development of CHC profiles of individual workers is associated with the regulation of CHC biosynthesis genes

Homogenization models for the development of colony-specific nestmate recognition cues predict that cue specificity is acquired by individuals via physical contact with other colony members and/or environmental sources of hydrocarbons (*Breed et al., 2015*; *Crozier and Dix, 1979*; *Lenoir et al., 2001*; *Meskali et al., 1995*; *Soroker et al., 1994*; *Soroker et al., 1995b*; *van Zweden et al., 2010*). However, because our data indicate that the maturation of the CHC profile of individual honey bees is actually regulated in association with the stereotypic age-dependent division of labor in this species, we next hypothesized that the CHC profiles of worker honey bees develop, at least in part, via an intrinsic age-dependent regulation of the CHC biosynthetic pathways in the pheromone producing oenocytes (*Chung and Carroll, 2015*; *Falcón et al., 2014*; *Makki et al., 2014*; *Yew and Chung, 2015*). Thus, we next examined whether age and/or task are associated with the mRNA expression levels of genes that encode elongases and desaturases, the primary CHC diversity producing classes

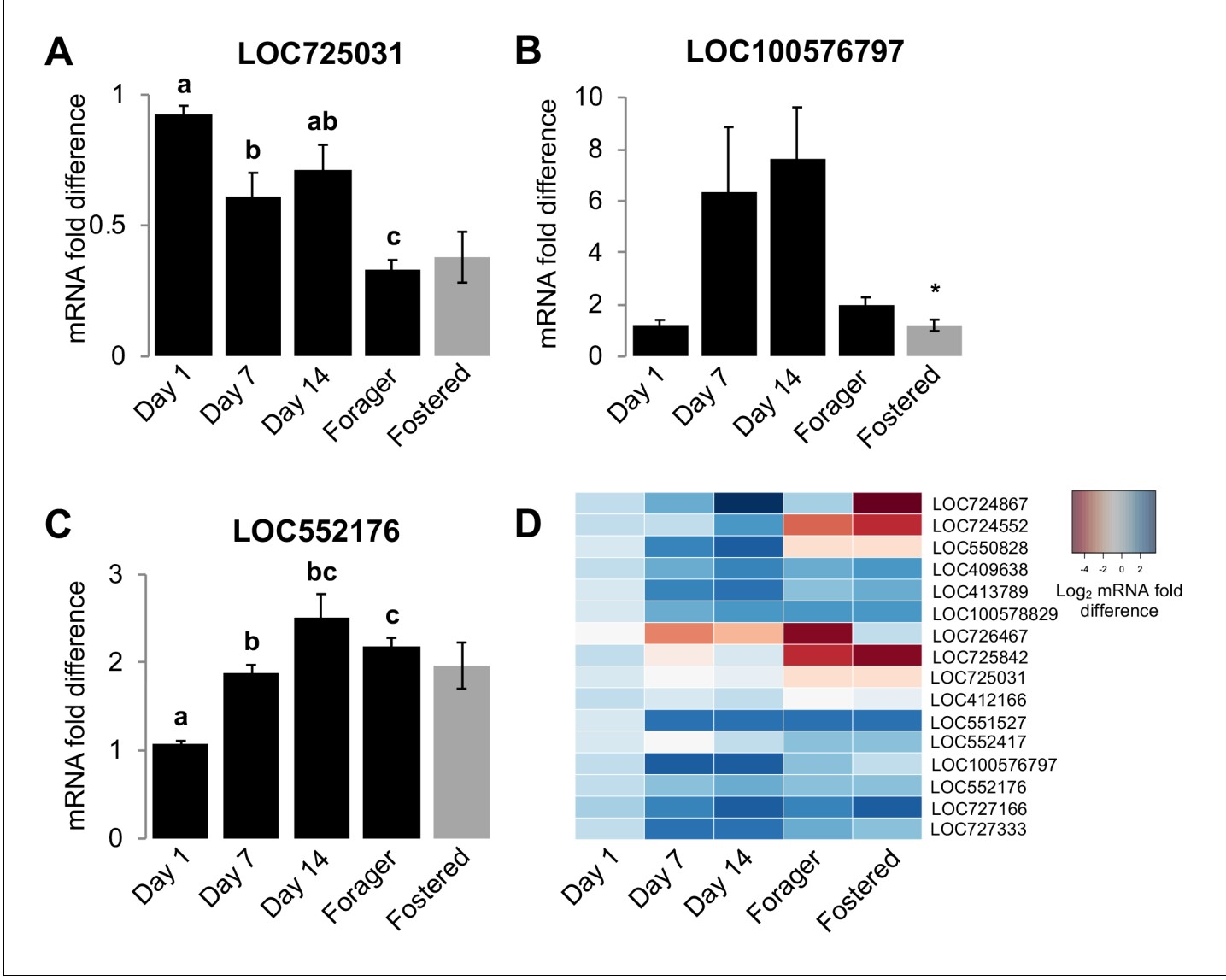

**Figure 4.** Age and social environment affect the expression level of CHC biosynthesis genes. (**A**) Elongase gene. (**B–C**) Desaturase genes. Only genes with different expression levels between at least two groups are shown (See *Table 7* for results for all studied genes). Black bars represent bees raised in their own colony. Grey bars represent sister forager bees that were raised in an unrelated colony ('Fostered'). (**D**) Heat map of relative expression levels of all genes tested. Aging bee statistics using ANOVA followed by Tukey HSD post-hoc, or Kruskal-Wallis followed by Dunn's Test with FDR adjustment post-hoc, with letters denoting *posthoc* statistical significance (p<0.05). Between colony statistics using Mann-Whitney U test, with asterisks above grey bars denoting statistical significance from foraging bees raised in their own colony (*, p<0.05). Sample size per group, N = 4.
DOI: https://doi.org/10.7554/eLife.41855.017

The following source data is available for figure 4:

**Source data 1.** Average $C_t$ scores across three technical replicates for each bee sample for every elongase and desaturase gene, including those corresponding to *Figure 4*.
DOI: https://doi.org/10.7554/eLife.41855.018

of enzymes in the CHC biosynthesis pathway (*Chung and Carroll, 2015*). To identify candidate genes for our analyses, we first used a bioinformatic approach to identify all putative members of both protein families in the honey bee genome (*Table 6*). Subsequently, we used real-time quantitative RT-PCR to compare mRNA levels of each candidate gene in dissected abdominal cuticles from bees of different ages raised in their source colony (sisters of the bees analyzed for *Figure 1- Figure 1—figure supplement 1*), as well as foraging sister bees raised in either their source colony or

**Table 7.** Genes differ in relative mRNA expression level between bees of different ages (Age), and foraging sister bees raised in two different colonies (Hive).

Numbers represent mean relative mRNA expression level ±standard error across four biological replicates. All p-values are from parametric ANOVA or nonparametric Kruskal Wallis ANOVA (denoted by 'KW'). Letters denote statistically significant age groups across individual compounds via Tukey's HSD (ANOVA post-hoc) or Dunn's Test with FDR adjustment (KW post-hoc) (p<0.05).

| Gene | Day 1 | Day 7 | Day 14 | Forager | FDR adjusted p-value *Age* | Fostered | FDR adjusted p-value *Hive* |
|---|---|---|---|---|---|---|---|
| LOC724867 | 1.22 ± 0.19 | 2.41 ± 0.75 | 12.62 ± 7.23 | 1.58 ± 0.88 | 0.328 (KW) | 1.22 ± 0.18 | 1 (KW) |
| LOC724552 | 1.05 ± 0.26 | 1.28 ± 0.66 | 2.94 ± 1.65 | 0.09 ± 0.02 | 0.177 (KW) | 0.03 ± 0.01 | 0.456 |
| LOC550828 | 1.02 ± 0.12 | 4.09 ± 1.99 | 6.17 ± 2.10 | 0.38 ± 0.09 | 0.097 (KW) | 0.30 ± 0.06 | 0.53 (KW) |
| LOC409638 | 1.14 ± 0.21 | 2.47 ± 0.41 | 4.36 ± 1.65 | 2.81 ± 0.26 | 0.097 (KW) | 4.07 ± 0.67 | 0.530 |
| LOC413789 | 1.03 ± 0.10 | 4.17 ± 0.68 | 5.31 ± 2.07 | 1.98 ± 0.15 | 0.056 (KW) | 2.38 ± 1.21 | 1 (KW) |
| LOC100578829 | 0.84 ± 0.08 | 2.30 ± 0.42 | 3.34 ± 0.71 | 3.12 ± 0.43 | 0.056 | 3.61 ± 2.25 | 0.53 (KW) |
| LOC726467 | 0.58 ± 0.18 | 0.12 ± 0.06 | 0.19 ± 0.06 | 0.03 ± 0.01 | 0.064 (KW) | 0.02 ± 0.00 | 0.53 (KW) |
| LOC725842 | 1.10 ± 0.27 | 0.46 ± 0.24 | 0.98 ± 0.47 | 0.05 ± 0.03 | 0.081 (KW) | 0.01 ± 0.00 | 0.53 (KW) |
| LOC725031 | 0.92 ± 0.04 (A) | 0.61 ± 0.09 (BC) | 0.71 ± 0.10 (AB) | 0.33 ± 0.04 (C) | 0.005 | 0.43 ± 0.10 | 0.573 |
| GB51249 | 1.16 ± 0.20 | 2.78 ± 0.44 | 4.98 ± 1.88 | 3.11 ± 0.25 | 0.097 (KW) | 4.14 ± 0.79 | 0.530 |
| LOC412166 | 1.06 ± 0.07 | 0.84 ± 0.12 | 1.08 ± 0.06 | 0.61 ± 0.05 | 0.081 (KW) | 0.76 ± 0.43 | 0.456 (KW) |
| LOC551527 | 0.92 ± 0.18 | 5.62 ± 1.08 | 4.89 ± 1.37 | 5.18 ± 0.82 | 0.056 | 4.99 ± 1.48 | 1 |
| LOC552417 | 0.93 ± 0.06 | 0.62 ± 0.07 | 1.19 ± 0.49 | 1.76 ± 0.97 | 0.352 (KW) | 1.76 ± 0.69 | 0.897 (KW) |
| LOC100576797 | 1.24 ± 0.17 | 6.38 ± 2.48 | 7.61 ± 2.01 | 1.96 ± 0.32 | 0.081 (KW) | 0.44 ± 0.22 (*) | 0.015 |
| LOC552176 | 1.08 ± 0.03 (A) | 1.87 ± 0.10 (B) | 2.51 ± 0.27 (B) | 2.18 ± 0.11 (B) | 0.003 | 1.76 ± 0.26 | 0.53 |
| LOC727166 | 1.42 ± 0.23 | 3.99 ± 0.67 | 6.41 ± 1.47 | 3.89 ± 0.77 | 0.056 | 8.58 ± 1.86 | 1 |
| LOC727333 | 1.16 ± 0.10 | 4.96 ± 1.44 | 5.11 ± 1.12 | 2.77 ± 0.91 | 0.081 | 1.95 ± 0.09 | 0.53 (KW) |

DOI: https://doi.org/10.7554/eLife.41855.020

an unrelated foster-colony. Our analyses revealed that the expression levels of at least one elongase and two desaturase genes are associated with either age or colony environment (*Figure 4* and *Table 7*). Thus, our data suggest that individual worker honey bees regulate CHC expression through an innate age-dependent developmental process that is further modulated by other factors such as task and the social environment.

## Age and task play a role in defining nestmate recognition cues in honey bee colonies

Because previous studies have indicated that nestmate recognition in honey bee colonies is likely driven by components of the CHCs profile (*van Zweden and D'Ettorre, 2010*), and our discovery that the CHC profiles of individual workers seem to mature in association with the well-described age-dependent division of labor in this species (*Robinson, 1992*; *Smith et al., 2008*; *Søvik et al., 2015*), we next hypothesized that, in honey bees, nestmate recognition cues themselves mature in association with age-dependent division of labor, and reach maturation during foraging. To test this hypothesis, we investigated the behavioral responses of guard bees to related and unrelated focal bees of different ages (Day 1, Day 7, Day 14, and foragers on Day 21). At each test colony, the behavioral responses of guards to random related and unrelated returning foragers of unknown age were used as the benchmark for the baseline level of nestmate recognition behavior. Behavioral observations revealed that bees are accepted at the entrance of their own colony, regardless of age (*Figure 5A*, Pearson's Chi-Squared, Day1: $\chi^2 = 49.05$, df = 2, p<0.001, Day 7: $\chi^2 = 19.07$, df = 2, p<0.001, Day 14: $\chi^2 = 44.89$, df = 2, p<0.001, Day 21: $\chi^2 = 28.32$, df = 2, p<0.001). In contrast, at the entrance to an unrelated colony, bees were accepted on Days 7 and 14, but rejected as foragers (Day 21) (*Figure 5B*, Day1: $\chi^2 = 11.61$, df = 2, p=0.003, Day 7: $\chi^2 = 15.51$, df = 2, p<0.001, Day 14: $\chi^2 = 11.91$, df = 2, p=0.002, Day 21: $\chi^2 = 7.35$, df = 2, p=0.04). These data support the hypothesis that nestmate recognition cues in honey bee colonies mature in association with age-dependent

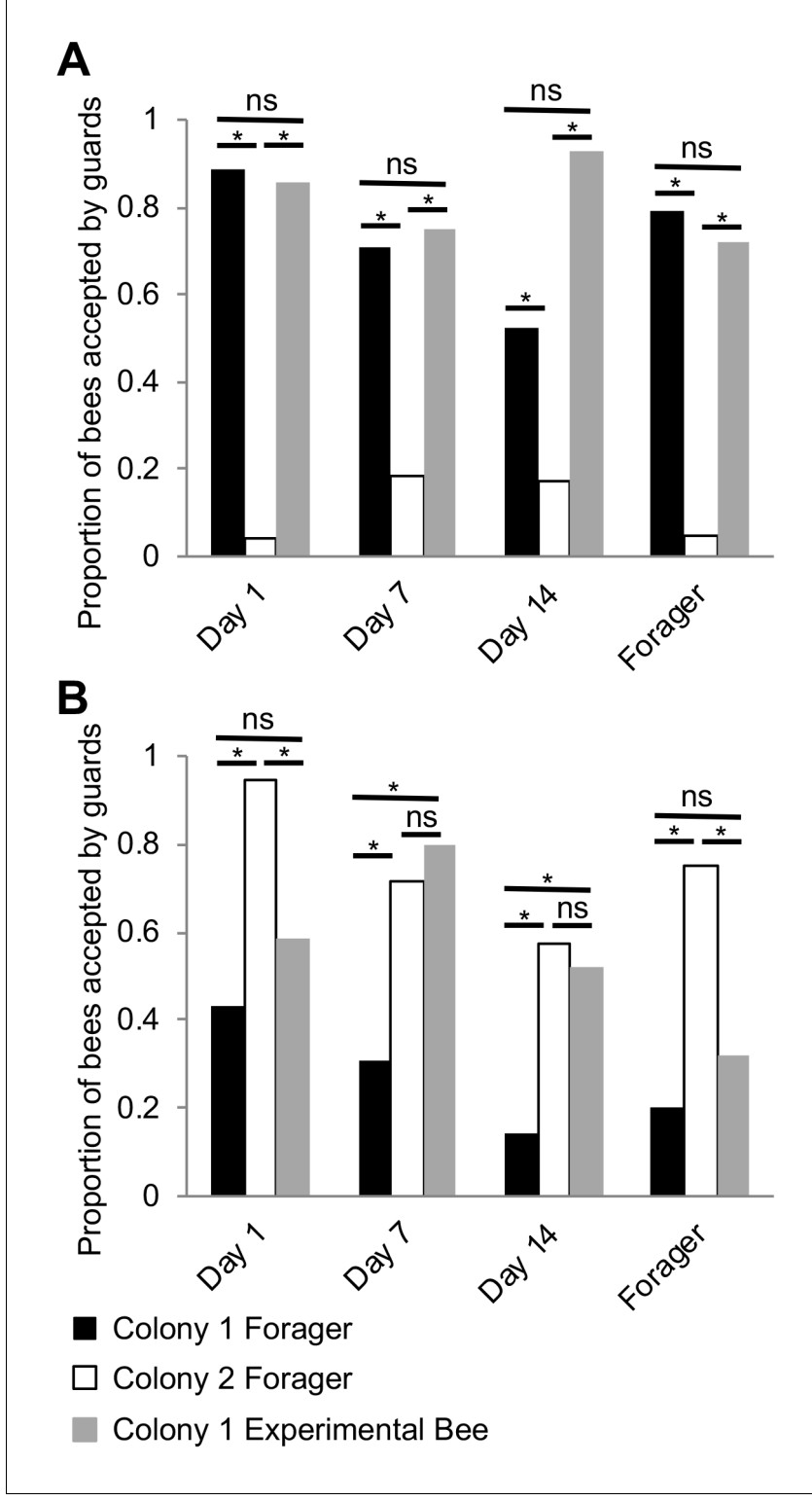

**Figure 5.** Nestmate recognition cues are forager-specific. (**A**) Bees are accepted at a similar rate as Colony one foragers at the entrance to their source colony (Colony 1) at all ages. (**B**) Bees are rejected at a similar rate as Colony one foragers at an unrelated colony (Colony 2) on Day one and Day 21. However, bees are accepted at a similar rate as Colony two foragers at an unrelated colony (Colony 2) on Day seven and Day 14. All statistics using Pearson's Chi-Square. Asterisks or letters denote *posthoc* statistical significance (p<0.05), ns denotes non-significant comparisons. Sample size per group, N = 18–29.

*Figure 5 continued on next page*

*Figure 5 continued*

DOI: https://doi.org/10.7554/eLife.41855.021

The following source data is available for figure 5:

**Source data 1.** Acceptance/Rejection scores for each bee tested in behavioral acceptance assays depicted in *Figure 5*.

DOI: https://doi.org/10.7554/eLife.41855.022

division of labor, and suggest that nestmate recognition is specific to behavioral interactions between guards and foragers at the entrance to the hive.

Surprisingly, we also observed that while young Day 1 bees are accepted by related guards, they are often rejected by unrelated guards (*Figure 5B*). This finding contradicts the broadly accepted 'blank slate' hypothesis, which predicts that because day-old bees are devoid of any defining chemical signatures, they should be always accepted by guards independent of relatedness (*Breed et al., 2004*). While we do not yet know which specific components of the CHC profile of young bees, if any, might have triggered a rejection by unrelated guards in our colonies, one plausible interpretation of these data is that the observed response of guards to unrelated Day 1 bees is an artifactual experimental outcome of a forced behavioral interaction between two bee groups, which in colonies with a typical demography, do not normally interact in the context of the hive entrance.

## Discussion

The ability of colonies of social insects to reliably recognize group membership is one of the remarkable adaptations that enabled their immense ecological success. Yet, the molecular and physiological mechanisms that underlie this complex trait remain unknown for most species. In the well-studied honey bee, previous studies suggested that the chemical cues that drive nestmate recognition are absent in newly eclosed bees, and subsequently develop primarily through the homogenization and transfer of chemicals between colony members via direct interactions such as allogrooming and trophallaxis, and indirect interactions such as physical contact with wax and other nest materials (*Breed et al., 2015*; *Breed et al., 2004*). However, the data we present here suggest that the overall development of individual CHC profiles of honey bee workers primarily depends on an innate developmental process that is associated with the stereotypic age-dependent division of labor in this species, and that colony-specific cues are likely only carried by foragers. Therefore, we posit that it is unlikely that CHC profiles in honey bees develop through homogenization and transfer mechanisms between nestmates and hive materials. Furthermore, given the established implicated role of CHCs in nestmate recognition (*van Zweden and D'Ettorre, 2010*), we additionally posit that CHC homogenization mechanisms are unlikely to play a key role in the production of colony-specific cues in honey bees.

A major line of investigation in understanding nestmate recognition of social insects has been to determine how colony-specific cues are determined. Cue specificity has historically been proposed to be determined by mechanisms under genetic control or acquired from the environment (*Crozier and Dix, 1979*). Although our studies do not directly address the mechanism by which cue specificity is determined in honey bees, data from cross-fostering experiments suggest that cue development and specificity are defined by interactions between factors derived from the colony-of-origin of individual workers and the actual hive environment they develop in. Therefore, our data suggest that CHC profiles of honey bee workers develop via a biphasic process that is governed, at least in part, by the intrinsic physiology of individual workers, the specific behavioral tasks they are engaged in, and the hive environment they age in. In phase one, similar to other social insect species (*Soroker et al., 1995a*), the total CHC amount builds up, possibly to increase the resistance of workers to desiccation while still inside the protective hive environment (*Chung and Carroll, 2015*). In phase two, the total amount of CHCs remains constant but the relative abundances of individual components shift in association with the age-dependent behavioral maturation of workers, at least in part, via the transcriptional regulation of CHC biosynthetic enzymes.

Which specific components of the honey bee CHC profile represent the nestmate recognition cue remains unknown. Although it has been shown that CHCs are likely used for nestmate recognition in

honey bees (*van Zweden and D'Ettorre, 2010*), it is unlikely that all components of the CHC profile contribute to this process (*Akino et al., 2004*; *Dani et al., 2005*; *Dani et al., 2001*; *Martin et al., 2008*; *Ruther et al., 2002*). In fact, it has previously been shown that alkenes seem to play a more prominent role in nestmate recognition in the honey bee than alkanes (*Dani et al., 2005*). Our data also indicate that although unrelated foragers raised in the same colony are equally accepted, their overall CHC profiles remain somewhat qualitatively different (*Figure 3C*). These data provide two important insights. First, guards are not likely using the full CHC profile of individuals to determine group membership. Second, differences in the CHC profiles of co-fostered nestmate foragers of similar age that originated from different source colonies indicate that the chemical profiles of individual workers are not likely to be the product of a stochastic CHC homogenization and transfer between colony members.

The observation that the mRNA expression levels of genes that encode CHC-biosynthesis enzymes vary in association with age and/or task further indicate that the primary mechanism for the dynamic regulation of the CHC profile of individual honey bee workers is directly associated with the well-established age-dependent division of labor in the honey bee (*Robinson, 1992*; *Smith et al., 2008*; *Søvik et al., 2015*). Although these data do not directly exclude the possibility that some particular CHCs are transferred across colony members, they do indicate that the overall observed qualitative age- and task-dependent changes in the CHC profiles of individual workers are affected by intrinsic molecular dynamics of the CHC synthesis pathway. However, our studies also importantly show that genetically-related bees that age in different colonies exhibit qualitatively different CHC profiles and CHC biosynthesis gene expression levels, which suggests that the CHC synthesis process is also plastic and could be modulated by factors associated with the hive/social environment.

We were initially surprised by our observation that Day one bees are accepted at the entrance to their source colony but rejected by guards at the entrance of an unrelated colony since previous studies hypothesized that the lower amounts of total CHCs in young bees represent a 'blank slate' in terms of the nestmate recognition cue because these bees are readily 'accepted' when introduced into unrelated colonies (*Breed et al., 2004*). In fact, this phenomenon was exploited here to introduce cohorts of bees to foster colonies, typically by placing the new bees on the top frames of experimental hives. This apparent conundrum highlights an important, yet often underappreciated, aspect of the nestmate recognition system in honey bees and other social insect species, which is that the 'rejection' behavior by guards is highly contextual. Conceptually analogous to other biological systems responsible for the detection of 'self' versus 'non-self' (*e.g.*, the acquired immunity system in vertebrates), behaviors associated with nestmate recognition are restricted to interactions between guards and incoming bees at the entrance to the hive (*Couvillon et al., 2013*). Therefore, we speculate that because nestmate recognition is spatially restricted to specific behavioral interactions between entering bees and guards at the entrance, the commonly observed 'acceptance' of day old bees outside the specific context of the hive entrance actually represents the lack of behavioral 'rejection' rather than a true self-recognition-dependent 'acceptance'. Consequently, the observation that Day one bees are rejected at the entrance of an unrelated colony suggests that nestmate recognition of young bees either depends on components of the CHC profile that are already present in Day one bees, non-CHC chemical cues, or an altogether different sensory modality. Alternatively, because newly eclosed bees usually perform cell cleaning behaviors at the interior of the hive, and therefore do not typically interact with guards at the hive entrance (*Robinson, 1992*; *Smith et al., 2008*; *Søvik et al., 2015*), differences in rejection of Day one bees between these two colonies might represent an experimental artifact resulting from differences in tolerance to the forced behavioral interaction between two bee groups that normally do not interact. Additionally, it has previously been shown that observed levels of guarding behaviors in honey bees are plastic, and could fluctuate in response to various environmental factors such as seasonal weather patterns, overall colony size, food availability, and 'robbing' pressures from other colonies or predators (*Downs and Ratnieks, 2000*). Likewise, more extreme forms of plasticity in nestmate recognition systems have been reported in other social species. For example, some social insects can switch between using visual or chemosensory modalities for nestmate recognition under different circumstances (*Baracchi et al., 2015*). Together, it seems that instead of being driven by simple binary decisions, nestmate recognition systems in the honey bee and other social insect species depend on a plastic recognition of 'friends' versus 'foes' as part of a broader group-level optimization of colony fitness.

In conclusion, we propose that nestmate recognition cue production and acquisition in honey bees are not likely to be primarily driven by CHC homogenization and transfer mechanisms as previously described in some ant species (*Boulay et al., 2000*; *Lenoir et al., 2001*; *Meskali et al., 1995*; *Soroker et al., 1994*; *Soroker et al., 1995b*; *van Zweden et al., 2010*). Instead, we propose a new model for the regulation of nestmate recognition in honey bee colonies, which stipulates that unknown factors associated with the hive environment play a direct or indirect role in defining the developmental kinetics and specificity of nestmate recognition cues by modulating the cellular and molecular processes that are responsible for pheromone synthesis. Thus, it is plausible that the colony/social environment drives the intrinsic development of similar pheromone profiles by individual colony members, which in typical honey bee hives, is associated with the physiological processes that drive age-dependent division of labor. If true, this model could resolve previous seemingly contradictory data which suggested that honey bee CHC profiles are defined by genetic (*Page et al., 1991*) versus environmental (*Downs and Ratnieks, 1999*) factors, as well as open the door for comparative mechanistic studies of how complex social traits evolve and function in different social insect clades.

## Materials and methods

### Animal husbandry and bee collections

Honey bee (*Apis mellifera*) colonies were reared and managed using standard beekeeping techniques across two locations near St. Louis, MO: Tyson Research center (38° 31′N, 90° 33′W) and a residential home. For all experiments that included collections of bees at specific ages, capped brood frames were taken from a colony and placed in a humidified 32°C incubator. Once eclosed, about 1000 bees (<24 hr old) were marked with a spot of paint (Testors, Vernon Hills, IL, USA) on their thorax, and then reintroduced into either their source or a foster colony, depending upon the experiment. For collections of bees at specific ages, marked bees were collected from internal frames of the colony one day post reintroduction (Day 1), seven days post reintroduction (Day 7), 14 days (Day 14) post reintroduction, and as returning foragers, identified by pollen loads on their hind legs or having a distended abdomen due to nectar loads, between 18 and 21 days post reintroduction. Bees used for chemical and molecular analyses were placed in individual 1.7 mL microtubes and immediately placed on dry ice. All samples were kept at −80°C until further analysis.

### Single-cohort colonies

Single-cohort colonies (SCC) were established as previously reported (*Ben-Shahar et al., 2004*; *Ben-Shahar et al., 2002*; *Greenberg et al., 2012*; *Robinson et al., 1989*; *Whitfield et al., 2003*). In short, about 1000 newly eclosed bees (<24 hr old) were placed in a small wooden nucleus hive-box with a young, unrelated mated queen, one honey frame from their source colony, an empty comb frame, and three new frames with wax covered plastic foundation. Bees were collected as typical-aged nurses and precocious foragers one week after introduction, and as over-aged nurses and typical-aged foragers at three weeks after introduction. Bee samples were collected and stored as above.

### Undertaker collection

To induce 'undertaking' behavior, about 1000 dead bees were placed into the top of two different colonies, and the first 20 bees that were observed removing dead bees from the colony were collected from the entrance. Returning foragers and in-hive nurses of unknown ages were also collected from each colony at the same time. Samples were stored and processed as described above.

### Big-back colony

Big-back colonies were established as previously described (*Ben-Shahar et al., 2000*; *Withers et al., 1995*). In short, bees were introduced in two cohorts to a 5-frame hive box containing three empty comb frames, two brood frames, and a new queen. In the first cohort, 200 day-old bees were collected as described above and marked on the thorax with paint. Half of these bees were marked with a plastic tag attached to the thorax (~3 mm diameter,~1 mm thick; 'big-back' bees). Day-old bees in the other cohort were collected and introduced 4 days later as described above to increase

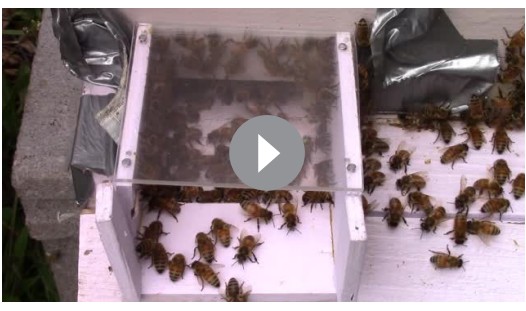

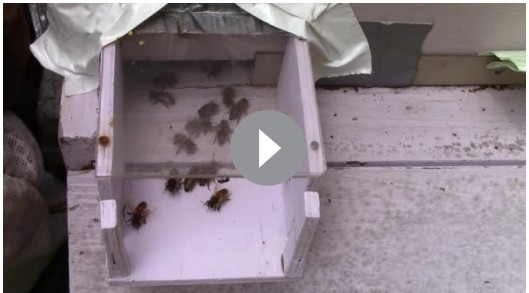

**Video 1.** An interaction between a guard and focal bee scored as 'Rejected'. The focal bee is marked with a green dot on its thorax.
DOI: https://doi.org/10.7554/eLife.41855.023

**Video 2.** An interaction between a guard and focal bee scored as 'Accepted'. The focal bee is marked with a pink dot on its thorax.
DOI: https://doi.org/10.7554/eLife.41855.024

the proportion of precocious foragers in the first group. The entrance to the colony was blocked by a piece of Plexiglas with holes in it that prevented 'big-back' bees from leaving the hive, but allowed paint marked bees to leave. Bees were collected at 7 days of age: returning foragers were collected as described above, and 'big-back' bees were collected as they were attempting to leave the hive via the holes in the plastic.

## Reversion colony

Reversion colonies were made by collecting ~1000 foragers from a single source colony by vacuuming them directly into a sealed 5-frame hive box containing two brood frames, one honey frame, and two empty comb frames. The hive was sealed and moved to a new location ~30 miles away from the source colony, and a new queen was added that night. The hive was sealed for 3 days, and then was opened to allow normal foraging activity to resume. During this time, in the absence of nurses, some foragers reverted back to nursing behaviors (*Robinson et al., 1992*). Actively foraging bees were collected at the hive entrance as described above and reverted nurses were collected from internal frames as described above.

## Cross-fostering experiment

1000 day-old bees from two independent source colonies were collected and marked as above. Half of the bees in each marked cohort were randomly reintroduced to both their own source colony and the reciprocal foster colony. Subsequently, marked bees of defined age were recollected from internal frames of each colony as described above.

## Nestmate recognition assay

Every day over a three-week period, newly eclosed bees (<24 hr old) from a single source colony were collected as described above, uniquely color-marked, and then reintroduced into their source colony. Subsequently, on each experimental day, bees from the following groups were collected, placed in individual 15 mL plastic tubes (Corning, Corning, NY, USA), and chilled on wet ice in an ice cooler up to 10 min before the assay in order to limit heat related stress: bees of the focal age (identified by color of mark), returning nectar foragers (denoted by distended abdomen and lack of pollen) of unknown age from the source colony, and returning nectar foragers of unknown age from an unrelated colony. All foragers, which served as behavioral controls, were painted the same color as the experimental bees just after collection. Tubes were numbered in a randomized order and blinded to the experimenter conducting the behavioral assays. Fifteen bees per group were prepared for each colony each experimental day.

Behavioral assays were conducted simultaneously at two colonies (source and unrelated) by two researchers, as well as recorded using digital video cameras. As described previously (*D'ettorre et al., 2006*; *Downs and Ratnieks, 2000*), acceptance at the colony entrance was used as a proxy for nestmate recognition by placing individual bees on a modified entrance platform and recording the behavioral reactions of guard bees for ~5 min. Bees were considered 'Rejected' if they

were bit, stung and/or dragged by at least one guard bee (*Video 1*). Bees were considered 'Accepted' if they were approached by guards, antennated and/or licked and then left alone (not bit), if they immediately entered the colony and were not removed by other bees, or if they remained on the platform and did not receive aggression (*Video 2*). After 5 min, focal bees that remained on the platform outside the colony were removed before the next assay. All behaviors were scored in real time, and videos were retained as back-up. All behavioral assays were conducted during a period of 10 days, between 12 and 4pm, with two days focusing on each age of experimental bee (N = 20–30 bees per group).

## Cuticular hydrocarbon extractions and GC analysis

CHCs were extracted from whole bees by placing individual bees into 6 mL glass vials fitted with 16 mm PTFE/silica septa screw caps (Agilent Crosslab, Santa Clara, CA, USA). Bee CHCs were extracted in 500 µL hexane containing 10 ng/µl of octadecane ($C_{18}$) and 10 ng/µl of hexacosane ($C_{26}$) (Millipore Sigma, St. Louis, MO), which served as injection standards. To achieve efficient extraction, each vial was gently agitated by vortexing (Fisher Scientific, Waltham, MA, USA) for 2 min at minimum speed. Extracts were immediately transferred to new 2 mL glass vials fitted with 9 mm PTFE lined caps (Agilent Crosslab, Santa Clara, CA, USA). In cases where experiments involved forager honey bees, all bees (including non-foragers) had their hind legs removed prior to extraction, in order to ensure removal of pollen. 100 µL of each extract was transferred to a new 2 mL glass vial and stored at −20°C for further analysis; the remaining 400 µL was stored at −80°C as back-up.

Representative pooled samples of foragers and nurses of known age were first analyzed by combined gas chromatography/mass spectrometry (GC/MS) for compound identification. Samples were run from $150^0$ (3 min hold) to $300^0$ at $5^0$/min. Compounds were identified by their fragmentation pattern as compared to synthetic compounds. For profile characterizations of individual bees, samples were analyzed using an Agilent 7890A gas chromatograph system with a flame ionization detector (GC/FID) and PTV injector (cool-on-column mode), and outfitted with a DB-1 20 m x 0.18 mm Agilent 121–1022 fused silica capillary column (Agilent Technologies, IncSanta Clara, CA, USA). Sample volumes of 1.0 µl were injected onto the column. Helium was the carrier gas and applied at a constant flow rate of 1 ml/min. Analysis of the extract was carried out with a column temperature profile that began at 50C (held for 1 min) and was ramped at 36.6 °C/min to 150C and then at 5 C/min to 280C, where it was held for 10 min. The injector and FID temperatures were programmed to 280C and 300C, respectively. Agilent OpenLAB CDS (EZChrom Edition) software was used to calculate the retention time and total area of each peak. Data were normalized to known quantity (ng) of internal standard hexacosane and all ng data are listed in source data.

## CHC biosynthesis gene identification, RNA Isolation and Quantitative Real-Time PCR

Members of the highly conserved desaturase and elongase gene families were identified in the honey bee genome by using the protein BLAST search tool (https://blast.ncbi.nlm.nih.gov/Blast.cgi) with annotated *Drosophila melanogaster* amino acid sequences (https://flybase.org) of elongase and desaturase genes known to play a role in CHC biosynthesis (*Chung and Carroll, 2015*). Initial homologs in the honey bee genome were chosen by picking the top match (highest total score and query cover, lowest E value) for each *D. melanogaster* gene, and possible paralogs of these putative genes were identified by subsequently using the NCBI protein BLAST tool (RRID:SCR_004870) with these genes' amino acid sequences. Many of these putative elongase and desaturase genes have previously been identified as possible CHC biosynthesis pathway genes in the honey bee (*Falcón et al., 2014*). E-values from the BLAST scans of the honey bee genome by using three canonical *Drosophila melanogaster* CHC biosynthesis genes, *EloF* (elongase subfamily), *Elo68α* (elongase subfamily), and *desat1* (fatty acid desaturase subfamily), are listed in *Table 6*.

To measure mRNA levels of individual genes, the cuticles from the abdomens of four bees per group were dissected out, and total RNA was extracted using the Trizol Reagent (Life Technologies, Grand Island, NY, USA). SuperScript II (Life Technologies, Grand Island, NY, USA) reverse transcriptase was used to generate cDNA templates from 500 ng of total RNA per sample by using random hexamers. A Bio-Rad (Hercules, CA, USA) CFX Connect Real-Time PCR Detection System and Bio-Rad iTaq Universal SYBR Green Supermix were subsequently used for estimating relative differences

in mRNA levels across samples (N = 4 per group, run in triplicate technical replications). Expression levels of the *EIF3-S8,* a housekeeping gene that has previously been used as a reference gene in honey bee studies of gene expression by us and others (*Alaux et al., 2009*; *Fischer and Grozinger, 2008*; *Greenberg et al., 2012*; *Mao et al., 2015*; *Richard et al., 2008*), was used as a loading control. To further ensure that the reported expression data for the experimental genes are robust, we first confirmed that the raw *EIF3-S8* $C_t$ values per total RNA used in the individual RT reactions were not affected by any of the studied groups included in our current study (Kruskal-Wallis, H = 3.299, df = 4, p=0.5091). $C_t$ data is listed in *Figure 4—source data 1*. The specific RT-PCR primers for each gene-specific assay are listed in *Table 6*.

## Statistical analysis

All CHC analyses included a set of 19 peaks that represent well-established honey bee CHCs, identified by comparing GC traces to published data (*Kather et al., 2011*). For the comparisons of total CHCs across groups (as in *Figure 1A*), total ng of all identified CHCs in each bee were analyzed using ANOVA followed by Tukey's HSD in R 3.3.2 (*Team, 2016*). For the remainder of the datasets, the relative proportion of each compound in each sample was calculated and then used in further statistical analysis. For each dataset, a permutation MANOVA was run using the ADONIS function in the vegan package of R (RRID:SCR_011950) with Bray-Curtis dissimilarity measures (*Oksanen et al., 2017*). Pairwise comparisons with FDR p-value correction were subsequently run on experiments where more than two groups were compared. Data were visualized using non-metric multidimensional scaling (metaMDS function in the vegan package of R (RRID:SCR_011950) (*Oksanen et al., 2017*)) using Bray-Curtis dissimilarity, and either 2 or three dimensions in order to minimize stress to <0.1. For *Table 1*, *Table 2*, *Table 3*, and *Table 4* an ANOVA followed by Tukey's HSD post-hoc comparison, or Kruskal-Wallis followed by Dunn's Test with FDR adjustment was performed using total ng (*Table 1* and *Table 2*) or proportions (*Table 3* and *Table 4* of each compound across bees of the four time point collections. For cross-fostering studies, power was assessed by performing pseudo multivariate dissimilarity-based standard error, a method for assessing sample-size adequacy in multivariate data, as described in and using code from *Anderson and Santana-Garcon (2015)*. For behavioral data, the proportion of bees accepted by guard honey bees was calculated for each experimental group at each colony at each day of age. A Pearson's chi-square was run for each day of age at each colony with subsequent pairwise comparisons. For qPCR data, relative expression levels were calculated as previously described (*Greenberg et al., 2012*; *Hill et al., 2017*; *Zheng et al., 2014*), using *eIF3-S8* as a loading control. Fold-expression data were generated by using the $2^{-\Delta\Delta CT}$ method (*Livak and Schmittgen, 2001*) and designating a single individual from the 'Day 1' group (*Figure 5*) as a calibrator. For statistical analyses, the $2^{-\Delta\Delta CT}$ scores were compared within each gene across bees of different groups using an ANOVA followed by Tukey's HSD post-hoc comparison, or Kruskal-Wallis followed by Dunn's Test with FDR adjustment. Overall test p-values were then adjusted using FDR correction to account for 16 independent comparisons (*Benjamini and Hochberg, 1995*).

## Acknowledgements

We thank Tyson Research Center faculty and staff and Ellen Hartz for beekeeping assistance. We thank Anthony Cantu and Brice Henson for assistance in bee collections, Iris Chin for assistance with dissections and RNA extractions, and Kyle Skottke for assistance with hive construction.

# Additional information

## Funding

| Funder | Grant reference number | Author |
|---|---|---|
| National Science Foundation | 1545778 | Yehuda Ben-Shahar |
| National Science Foundation | 1707221 | Yehuda Ben-Shahar |
| National Science Foundation | 1754264 | Yehuda Ben-Shahar |

| | |
|---|---|
| Natural Sciences and Engineering Research Council of Canada | Joel D Levine |
| Canadian Institutes of Health Research | Joshua J Krupp<br>Joel D Levine |

The funders had no role in study design, data collection and interpretation, or the decision to submit the work for publication.

## Author contributions

Cassondra L Vernier, Conceptualization, Data curation, Formal analysis, Validation, Investigation, Methodology, Writing—original draft, Writing—review and editing; Joshua J Krupp, Formal analysis, Validation, Writing—review and editing; Katelyn Marcus, Data curation, Validation, Writing—review and editing; Abraham Hefetz, Formal analysis, Methodology, Writing—review and editing; Joel D Levine, Resources, Formal analysis, Validation, Methodology, Writing—review and editing; Yehuda Ben-Shahar, Conceptualization, Formal analysis, Supervision, Funding acquisition, Investigation, Writing—original draft, Project administration, Writing—review and editing

## Author ORCIDs

Cassondra L Vernier http://orcid.org/0000-0002-7381-3833
Abraham Hefetz http://orcid.org/0000-0001-9678-9429
Joel D Levine http://orcid.org/0000-0002-6254-6274
Yehuda Ben-Shahar http://orcid.org/0000-0002-2956-2926

## Decision letter and Author response

Decision letter https://doi.org/10.7554/eLife.41855.027
Author response https://doi.org/10.7554/eLife.41855.028

## Additional files

### Supplementary files

• Transparent reporting form
DOI: https://doi.org/10.7554/eLife.41855.025

### Data availability

All data generated or analysed during this study are included in the manuscript and supporting files. All CHC chemical data are included in the data source files.

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
