## [Decision Letter]

Thank you for submitting your article "Chemical signatures of honey bee group membership develop via a socially-modulated innate process" for consideration by *eLife*. Your article has been reviewed by three peer reviewers, and the evaluation has been overseen by Kristin Scott as the Reviewing Editor and Catherine Dulac as the Senior Editor. The reviewers have opted to remain anonymous.

The reviewers have discussed the reviews with one another and the Reviewing Editor has drafted this decision to help you prepare a revised submission.

Summary:

The ability to tell nestmates from non-nestmates is a central feature in the biology of social insects. Even though substantial effort has been dedicated to identifying the underlying chemical signatures, this process is still rather poorly understood. The current study presents a detailed analysis of the diversity and temporal dynamics of cuticular hydrocarbon (CHC) profiles, which are thought to be the main chemical cues mediating non-nestmate discrimination, in the honeybee. CHCs were examined across bees of different ages and behavioral castes, gene expression was quantified for genes involved in CHC synthesis, and behavioral experiments were performed to assay nestmate recognition behavior. The experiments and analyses are well-conceived and well-executed, and the results are generally clear, with some concerns to be addressed. Overall, the authors make a compelling case that several different factors are important in determining the CHC profile of individual bees, and that a model of simple CHC sharing among individual colony members is an oversimplification.

Essential revisions:

1) A significant shortcoming of this manuscript is that we don't know which of the CHCs studied (if any) are actually used for nestmate recognition. Probably some of them are, maybe even many of them, but CHCs also have other functions that are unrelated to nestmate recognition (e.g. barriers to desiccation, microbial infection, and abrasion). The authors should be very clear throughout the paper that they are studying entire CHC profiles, which is not the same thing as studying the CHCs used for nestmate recognition.

2) While the authors' evidence argues against simple Gestalt-style odor sharing, this really seems like a straw man. Many current researchers of social insect chemical ecology do not still subscribe to this view, especially not in the strict sense presented here. Current models for the dynamic of labels in nestmate recognition are much more nuanced, and there are numerous studies (in ants, bees, and wasps) showing that nestmates of different types (different castes, different tasks, different ages, social parasites) can have different cuticular hydrocarbon profiles. The Introduction and Discussion oversimplify the data on ants, what is known about colony "Gestalt" odors, and what exactly the concept of a "Gestalt" odor is. Several key papers are not cited and discussed. This gives off the impression that the current data on honeybees imply that bees are fundamentally different from ants, when in fact several studies on ants have suggested similar dynamics. For example, it has been shown that age (Cuvillier-Hot et al., 2001) and worker "subcaste"/task group (Wagner et al., 1998; Martin and Drijfhout 2009) can be an important determinant of CHCs in ants, i.e. not all workers in a colony necessarily display the same CHC profile. Therefore, as the authors show here for honeybees, there is evidence that at least in some ants the individual CHC profile changes as a function of age and age-related task differences. But that doesn't necessarily contradict the concept of a "Gestalt" odor component in the sense that workers from the same colony still smell sufficiently similar to each other (e.g. via genetic similarity, allogrooming, shared nest material and food sources, etc.) and thus can be distinguished from non-nestmates. An illustration of this idea can e.g. be found in Figure 2 of Sturgis and Gordon, 2012 (a review that seems very relevant to the current paper).

Please thoroughly revise the manuscript in light of a more careful consideration of the social insect literature and qualify statements about how the current findings contradict the concept of a "Gestalt" odor in nestmate recognition. Examples are in the Introduction paragraph five, Results first and fourth paragraph, (I don't see the "potential conundrum"), fifth paragraph (but this also doesn't mean that these factors are not important, i.e. it could be a mix of the two – a Gestalt component interacting with an individual component), and then of course throughout the Discussion section. I'm listing these and a few other additional studies that seem to be very relevant to the claims of the current manuscript but are not currently discussed below.

A few additional specific thoughts on CHCs and nestmate recognition: Introduction paragraph one: individuals usually do not recognize relatedness per se (in fact I don't know of any case of strict kin-recognition in social insects), but use some proxy of relatedness (spatial proximity, colony odor, etc.); paragraph three: I'm not sure this really is the consensus in the field. My impression is that genetics does play an important role in this context, but that other factors (like the ones you list below) are important as well. The Gestalt odor then integrates these different factors over the individuals in a colony; Also in paragraph three: why "passive"? Many of the factors thought to contribute to this process are active (allogrooming, trophalaxis…). For example, queens of social parasites often actively groom and chew on resident workers before entering a host colony. This seems to be an adaptive behavior specifically aimed at achieving chemical camouflage; Again in paragraph three: I'm not aware of ant species that do not display allogrooming behavior (can you provide a reference?); Paragraph four: maybe mention above that the "blank slate" idea also applies to ants?

3) Figure 1 and related passages: subsection “CHC profiles of individual honey bee workers exhibit qualitative and quantitative age-dependent changes”: There is no significant increase between days 1 and 7. I would reformulate this to "increases between one-day post-reintroduction and 14 days post-reintroduction and then remains stable"; I would also mention the post-hoc statistics showing which groups exactly are different from each other here. This is currently only given in the figure itself. It's kind of nice though, because the qualitative changes seem to broadly coincide with the quantitative changes in Figure 1A. Regarding this analysis and the data in Table 1, is there a technical reason that limits you to working with proportions? The way you analyze these data forces you to keep the change in overall amount separate from "qualitative", i.e. relative changes in proportion. But it seems that the picture would be much clearer if you could simply show which compounds change quantitatively with age, and which ones do not. As it is now, you might well recover compounds that change in relative proportion, but that are completely invariable in terms of quantity, while you might miss some that change significantly in quantity but remain relatively stable in overall proportion. It's straightforward to calculate proportions from the quantities of all compounds, but you can't do it the other way around. So I feel that presenting the data the former way would be more informative. It would also be helpful to use the same color codes for different compounds in the Figure 1C,D and Figure 1—figure supplement 1C,D panels.

4) Figure 2 and related passages: The environmental experience of undertakers and foragers is probably quite different, in contrast to the perspective of the authors. Unless the authors can determine the time spent outside for undertakes and foragers, it is plausible that differences in environmental exposure contribute to CHC profiles, putting a hole in the argument that environmental exposure is not important to the development of CHC profiles. This is something that the authors will have to grapple with in their revision. Additionally, in subsection “The CHC signatures of individual workers are task-related”: I find it confusing that you present significant p-values to back up a statement that something is not different. Also, how old are the undertakers? According to what you write above, they should differ as long as undertakers are less than seven days and foragers are more than seven days old. If you look at Figure 2D it indeed looks like the undertakers have less CHCs than foragers, even though the difference is not statistically significant. I would therefore be careful with how you phrase this here. Something like "we did not detect a statistically significant difference" might be better than "we found that.… undertakers do not differ from foragers". Such a negative finding would have to be backed up by a careful statistical power analysis and then refer to specific effect sizes etc. So to me it looks like undertakers are very similar to nurses both in terms of total amount and profile quality; Second paragraph, final sentence: Is this a valid comparison?

5) Figure 3 and related passages: subsection “The development of individual CHC profiles is a regulated process modulated by the colony environment”: Here again, I think this should be formulated more carefully. This section hinges on the conclusion that workers from the same colony don't differ at days 7 and 14, but they do at foraging onset. So for the important negative results the authors should present a careful power analysis to show that they would be able to detect the kinds of effect sizes observed between foragers if they were present at earlier time points. Looking at Figure 3, this is really far from clear – the data actually look quite messy in that there appears to be some separation in all three cases, but there is also always a lot of overlap. So you'd really have to convince the reader that C is in fact different from A and B. I also find the term "natal sisters" confusing. If I understand correctly, these are the sisters that were put back into the natal colony of the fostered individuals, but they are not natal to the colony the experimental bees were fostered into. I would reformulate this to avoid the possible confusion. Change "factors associated with the natal colony" to "genetics or other factors associated with the natal colony"?

6) Figure 4: This figure is not very informative and I would move it to the supplement. The trees are essentially not resolved – I would collapse all nodes with less than 50 percent bootstrap support – the displayed topology is quite misleading. Also, for ML analyses it's more informative to display the single ML tree with branch lengths and then map bootstrap support onto that tree (rather than presenting a consensus tree without branch length). Finally, how were the outgroups chosen and how can you be sure they're actual outgroups? I couldn't find any justification in the Materials and methods.

7) Figure 5 and related passages: The authors chose to conduct qPCR to measure gene expression, and they use a single reference gene based on previously published studies. The standard for these types of analyses would be to use several reference genes that are first validated in the specific context of a given study. Furthermore, there are quite a few genes being tested, and that should be taken into account in multiple-comparison corrections, i.e. p-values for significance cutoffs should be adjusted accordingly. Based on the current data I could probably be convinced that there is some dynamic gene regulation going on, but looking at Figure 5 and the associated supplementary table, I find it really hard to say which of the indicated changes are actually real and biologically meaningful and which ones are spurious. Also, given the availability of RNA-seq approaches, it seems surprising that the authors still opted for qPCRs – RNA-seq would be much broader in scope and the statistical analysis would be more robust. I therefore feel that this is somewhat of a weakness of this study, and if at all possible, I would collect RNA-seq data in addition to the qPCR data. Patterns that hold up across both datasets would then be quite convincing. Figure 5 and the legend: remove indications of p<0.1 – these are not significant.

8) The authors present data showing that CHCs differ across ages and tasks and, separately, that expression of elongases and desaturases differ across ages and tasks, and the authors argue that these are manifestations of the same underlying process. Thus, it seems important to test the relationship between these two. For example, since desaturases catalyze the formation of double bonds in alkenes, are there more alkenes produced by bees that express higher levels of desaturases?

9) Figure 6 and related passages: subsection “Age and task play a role in defining nestmate recognition cues in honey bee colonies.”: Shouldn't statistical comparisons be between natal foragers and other types of natal bees? It's not intuitive to say that they are accepted by showing comparisons that show significant differences. Figure 6A: Again, this is not very intuitive. What this panel is supposed to show is that bees of all ages are accepted into their own colony at the same rate as foragers from the same colony are. However, all the statistical comparisons are between natal bees and bees from a different colony, rather than between different types of natal bees and natal foragers. What this really shows is that, as expected, natal bees of any kind are accepted into their colony more readily than foreign foragers. If that's the message of this figure then I would present it as such. If, on the other hand, this is supposed to compare different types of natal bees to natal foragers as a baseline, then the statistics would have to be done differently. For example, it looks like 14 day old bees are much more readily accepted into the colony than foragers. If these statistical comparisons have in fact been done and they are all non-significant, then I would indicate this with bars labeled "ns" in the figure. It would also be helpful to have standard errors on the bars. First paragraph, penultimate sentence: Same problem here. All of these comparisons are statistically significant, so it's not obvious how they support the claim made here. These don't seem to be the most relevant statistical comparisons. Figure 6B: Similar problems here. Comparisons that are not significant should be indicated. Also in paragraph two: I'm not sure how this follows from the previous experiments and the last experiment (Figure 6C) doesn't seem particularly relevant to the rest of the study. The alarm pheromone experiment is also challenging as different doses may yield different results. Removing Figure 6C would strengthen the manuscript.

References:

Greene, M.J. and D.M. Gordon (2003) Cuticular hydrocarbons inform task decisions. Nature 423:32.

Greene, M.J., Pinter-Wollman, N., and D.M. Gordon (2013) Interactions with combined chemical cues inform harvester ant foragers' decisions to leave the nest in search of food. PLoS ONE 8(1):e52219.

[Editors' note: further revisions were requested prior to acceptance, as described below.]

Thank you for resubmitting your work entitled "Chemical signatures of honey bee group membership develop via a socially-modulated innate process" for further consideration at *eLife*. Your revised article has been favorably evaluated by Catherine Dulac (Senior Editor), Kristin Scott (Reviewing Editor), and one reviewer.

The manuscript has been improved but there are a few textual changes that need to be addressed before acceptance. Specifically, care should be taken to use the term CHCs instead of chemical signatures or recognition cues, as detailed below. The discussion of the Gestalt model and implications from this study should be more nuanced, as described below.

Reviewer #2:

The revised manuscript is much improved, especially the way the results are presented. There are still a couple of issues where I think the authors should present their findings more carefully though. Maybe most importantly, as was already noted in the first round of reviews, it is not known which cues honeybees actually use for non-nestmate discrimination. What the authors study here are the dynamic changes in cuticular hydrocarbon profiles, without knowledge of their actual biological function. It is of course reasonable to say in the Introduction that CHCs have been implicated in non-nestmate discrimination, and to later discuss the possible implications of their findings in that context, but I would leave it at that, and be very careful in how to phrase this. I think that would further strengthen the manuscript. For example, I would change the title from "Chemical signatures of honey bee group membership…" to "Honey bee cuticular hydrocarbon profiles…" – you really don't know whether the observed changes are signatures of group membership, at least in any biologically meaningful sense. If anything, the results in subsection “The development of individual CHC profiles is a regulated process modulated by the colony environment”, for example, suggest that they are not. Similarly, in the Abstract, I would replace all mentions of "nestmate recognition cues" with "cuticular hydrocarbon profiles". Same in Discussion paragraph one. Introduction final paragraph: you don't know what the "colony-specific nestmate recognition cues" are, so you can't say anything about that directly.

As an aside, and I know it sounds bulky, I prefer "non-nestmate discrimination" over "nestmate recognition". At a behavioral level, what is usually measured is that social insects discriminate against (attack) non-nestmates, not whether they recognize nestmates (you could make a similar argument for models of the underlying neuronal computation). I know that's not necessarily reflected in the literature and I would therefore leave it up to the authors, but in my opinion this is a better way to think about the behavior.

I still find it somewhat unfortunate how the study is set up in the Introduction, and how the findings are summarized. E.g. "Yet, whether "Gestalt"-like mechanisms regulate nestmate recognition systems in other social insect species remains mostly unexplored."; "However, whether "Gestalt"-like homogenization mechanisms play a role in honey bee nestmate recognition remains unknown."; and then the brief summary at the start of the final paragraph of the Introduction, saying that the authors' findings from honey bees contradict a Gestalt model. I don't think they do. You really don't know what the nestmate recognition cues are, and whether you're even capturing them in your analyses, so you can't exclude that the Gestalt model actually applies to these cues, and that the age-dependent changes you pick up here are either irrelevant (to the bees), or represent e.g. desiccation barriers and/or task group recognition cues that are superimposed on colony recognition cues (but have nothing to do with non-nestmate discrimination). Same problem Discussion final paragraph. Subsection “The CHC signatures of individual workers are task-related” paragraph three, sentence four: Doesn't this argue for some sort of Gestalt (or maybe genetic) effect?

---

## [Author Response]

Essential revisions:1) A significant shortcoming of this manuscript is that we don't know which of the CHCs studied (if any) are actually used for nestmate recognition. Probably some of them are, maybe even many of them, but CHCs also have other functions that are unrelated to nestmate recognition (e.g. barriers to desiccation, microbial infection, and abrasion). The authors should be very clear throughout the paper that they are studying entire CHC profiles, which is not the same thing as studying the CHCs used for nestmate recognition.

We thank the reviewers for bringing our attention to this important point. As described in details below, we now reemphasize in the Introduction, Results, and Discussion that our analyses and data interpretations primarily focused on the phenotypic plasticity of the overall CHC profile of individual bees, and that the chemical identity of the components that might be used specifically for nestmate recognition in honey bees remains unknown:

“Whether the overall profile, or more specific components of it, represent the actual nestmate recognition cue remains unknown. However, previous studies have indicated that variations in the relative amounts of each compound in the CHC profile across individuals from different colonies are likely sufficient for the chemical recognition of nest membership (van Zweden and D’Ettorre, 2010).”

“Because previous studies have indicated that nestmate recognition in honey bee colonies is likely driven by components of the CHCs profile (van Zweden and D’Ettorre, 2010), and our discovery that the CHC profiles of individual workers seem to mature in association with the well-described age-dependent division of labor in this species (Robinson, 1992; Smith et al., 2008; Søvik et al., 2015), we next hypothesized that, in honey bees, nestmate recognition cues themselves mature in association with age-dependent division of labor, and reach maturation during foraging.”

“Which specific components of the honey bee CHC profile represent the nestmate recognition cue remains unknown. Although it has been shown that CHCs are likely the main chemicals used for nestmate recognition in honey bees (van Zweden and D’Ettorre, 2010), it is unlikely that all components of the CHC profile contribute to this process (Akino et al., 2004; Dani et al., 2005, 2001; Martin et al., 2008; Ruther et al., 2002). In fact, it has previously been shown that alkenes seem to play a more prominent role in nestmate recognition in the honey bee than alkanes (Dani et al., 2005). Our data also indicate that although unrelated foragers raised in the same colony are equally accepted, their overall CHC profiles remain somewhat qualitatively different (Figure 3C). These data provide two important insights. First, guards are not likely using the full CHC profile of individuals to determine group membership. Second, differences in the CHC profiles of co-fostered nestmate foragers of similar age that originated from different source colonies indicate that the chemical profiles of individual workers are not likely to be the product of a stochastic CHC homogenization and transfer between colony members.”

2) While the authors' evidence argues against simple Gestalt-style odor sharing, this really seems like a straw man. Many current researchers of social insect chemical ecology do not still subscribe to this view, especially not in the strict sense presented here. Current models for the dynamic of labels in nestmate recognition are much more nuanced, and there are numerous studies (in ants, bees, and wasps) showing that nestmates of different types (different castes, different tasks, different ages, social parasites) can have different cuticular hydrocarbon profiles. The Introduction and Discussion oversimplify the data on ants, what is known about colony "Gestalt" odors, and what exactly the concept of a "Gestalt" odor is. Several key papers are not cited and discussed. This gives off the impression that the current data on honeybees imply that bees are fundamentally different from ants, when in fact several studies on ants have suggested similar dynamics. For example, it has been shown that age (Cuvillier-Hot et al., 2001) and worker "subcaste"/task group (Wagner et al., 1998; Martin and Drijfhout, 2009) can be an important determinant of CHCs in ants, i.e. not all workers in a colony necessarily display the same CHC profile. Therefore, as the authors show here for honeybees, there is evidence that at least in some ants the individual CHC profile changes as a function of age and age-related task differences. But that doesn't necessarily contradict the concept of a "Gestalt" odor component in the sense that workers from the same colony still smell sufficiently similar to each other (e.g. via genetic similarity, allogrooming, shared nest material and food sources, etc.) and thus can be distinguished from non-nestmates. An illustration of this idea can e.g. be found in Figure 2 of Sturgis and Gordon, 2012 (a review that seems very relevant to the current paper).

We agree with the reviewers that our original introduction to the Gestalt model was oversimplified. We also regret the omission of key papers related to it. Because the use and interpretation of what “Gestalt” means has been somewhat loose in different studies of nestmate recognition systems, we made a choice to interpret it as it was originally framed by Crozier and Dix (1979), and used by previous studies of nestmate recognition in honey bees. As we understand it in the context of the current study, the “Gestalt” model refers to the mechanism by which colony-specific cues are acquired by individual colony members (i.e. homogenization of chemicals primarily through contact with other colony members and hive materials) rather than the actual identity or sources of the cue, and therefore, here we interpreted our data specifically in this context. To address this important point, we revised the relevant sections in the Introduction as follows:

“Consequently, empirical and theoretical studies suggested that individual colony members acquire their colony-specific chemical signature via a cue homogenization process involving the exchange of relevant chemicals through interactions between colony members or contact with nest building materials, often referred to as the “Gestalt” model (Crozier and Dix, 1979). Empirical evidence in support of the “Gestalt” model have been reported for a few ant species, which are known to transfer mixed blends of CHCs between individuals through trophallaxis and grooming via the action of the postpharyngeal gland (PPG) (Boulay et al., 2000; Lenoir et al., 2001; Meskali et al., 1995; Soroker et al., 1994; Victoria Soroker et al., 1995; Van Zweden et al., 2010). However, other studies suggest that many of the predictions of the classic “Gestalt” model might not fully represent the processes that drive the development of chemical nestmate recognition cues in all social insect species. For example, some ant species do not display robust trophallaxis behaviors, the main mode of chemical transfer across colony members (Soroker et al., 1994; Soroker et al., 1995), and in others, the chemical profiles of individual colony members are likely modulated by genetic relatedness (Teseo et al., 2014), age (Cuvillier-Hot et al., 2001; Teseo et al., 2014), and/or task (Martin and Drijfhout, 2009; Sturgis et al., 2012; Wagner et al., 2001, 1998). Together, these data suggest that the regulation of chemical nestmate recognition cues in different species is more variable and complex than initially hypothesized (Esponda et al., 2015; Newey, 2011; Sturgis and Gordon, 2012).”

“Based on these studies, it has been hypothesized that, similar to some ant species, the CHC profile of newly eclosed workers represents a “blank slate” (Breed et al., 2004; Lenoir et al., 1999), and that, similar to predictions of the classic “Gestalt” model, nestmate recognition cues are subsequently acquired by individual workers via the homogenization and transfer of chemicals through direct social interactions and intermediate environmental factors (Breed et al., 2015). Furthermore, it has recently been proposed that the cephalic salivary gland of honey bee workers is functionally analogous to the PPG in ants, and is likely involved in the “Gestalt”-like homogenization and transfer of the chemical cues shared by all colony members (Martin et al., 2018). However, whether “Gestalt”-like homogenization mechanisms play a role in honey bee nestmate recognition remains unknown.”

Please thoroughly revise the manuscript in light of a more careful consideration of the social insect literature and qualify statements about how the current findings contradict the concept of a "Gestalt" odor in nestmate recognition. Examples are in the Introduction paragraph five, Results first and fourth paragraph, (I don't see the "potential conundrum"), fifth paragraph (but this also doesn't mean that these factors are not important, i.e. it could be a mix of the two – a Gestalt component interacting with an individual component), and then of course throughout the Discussion section. I'm listing these and a few other additional studies that seem to be very relevant to the claims of the current manuscript but are not currently discussed below.

If we understand this comment correctly, the reviewers refer to the term “Gestalt” to describe actual components of the chemical profile or “odor” that signals group membership, and thus interpret the concept of a "Gestalt" as an odor that is similar across nestmates. However, the concept of a nestmate recognition cue in itself implies that a similar odor (or chemical profile) is shared between nestmates, which we fully agree with. Therefore, as we describe above, here we interpreted “Gestalt”, and related concepts, according to their meaning in the original model, which uses “Gestalt” to the describe the physiological and behavioral processes that mediate the development and acquisition of colony-specific nestmate recognition cues by individual colony members (e.g., homogenization mechanisms), but not the specific chemical identities of the cues. Therefore, we respectfully argue that our claim that the data presented here do not support the predictions of the Gestalt model still holds. In addition to our response to the previous comment, we clarify this important point in the revised text as follows:

“Here we provide empirical evidence that, in contrast to predictions of the classic “Gestalt model”, which stipulates that nestmate recognition cues are primarily directly acquired from the environment, other group members, or hive building materials, the maturation of the pheromonal profile of individual honey bee workers is primarily regulated by innate developmental processes associated with age-dependent behavioral tasks and modulated by the social colony environment. Specifically, we find that individual workers exhibit stereotypic quantitative and qualitative changes in their CHC profile as they transition from in-hive tasks to foraging outside, that these changes are associated with innate transcriptional changes in CHC biosynthetic pathway genes, and that the foraging task is directly associated with the final maturation of the colony-specific nestmate recognition cue. Together, our findings suggest that not all members of honey bee colonies display a uniform chemical signature via the direct acquisition of CHCs. Instead, our data indicate that nestmate recognition cues in honey bees are more likely a product of a genetically-determined developmental program that is modulated by colony-specific factors.”

““Gestalt”-like models for the development of colony-specific nestmate recognition cues predict that cue specificity is acquired by individuals via physical contact with other colony members and/or environmental sources of hydrocarbons (Breed et al., 2015; Crozier and Dix, 1979; Lenoir et al., 2001; Meskali et al., 1995; Soroker et al., 1994; Victoria Soroker et al., 1995; Van Zweden et al., 2010). However, because our data indicate that the maturation of the CHC profile of individual honey bees is actually regulated in association with the stereotypic age-dependent division of labor in this species, we next hypothesized that the CHC profiles of worker honey bees develop, at least in part, via an intrinsic age-dependent regulation of the CHC biosynthetic pathways in the pheromone producing oenocytes (Chung and Carroll, 2015; Falcón et al., 2014; Makki et al., 2014; Yew and Chung, 2015).”

“Thus, in contrast to the previous assumption that the CHC profiles of worker bees are primarily a product of their interactions with other colony members and/or nest materials, our data suggest that individual worker honey bees regulate CHC expression through an innate age-dependent developmental process that is further modulated by other factors such as task and the social environment.”

“In the well-studied honey bee, previous studies suggested that the chemical cues that drive nestmate recognition develop via a “Gestalt”-like homogenization and transfer of chemicals between colony members through direct interactions such as allogrooming and trophallaxis, and indirect interactions such as physical contact with wax and other nest materials (Breed et al., 2015). However, the data we present here suggest that nestmate recognition cues in the honey bee primarily depend on an innate developmental process that is associated with the stereotypic age-dependent division of labor in this species, and do not completely mature until bees make their final behavioral transition to foraging.”

“In conclusion, we propose that nestmate recognition cue production and acquisition in honey bees is not likely to be driven by processes that are analogous to the “Gestalt”-like mechanisms as previously described in some ant species (Boulay et al., 2000; Lenoir et al., 2001; Meskali et al., 1995; Soroker et al., 1994; Victoria Soroker et al., 1995; Van Zweden et al., 2010). Instead, we propose a new model for the regulation of nestmate recognition in honey bee colonies, which stipulates that unknown factors associated with the hive environment play a direct or indirect role in defining the developmental kinetics and specificity of nestmate recognition cues by modulating the cellular and molecular processes that are responsible for pheromone synthesis.”

A few additional specific thoughts on CHCs and nestmate recognition: Introduction paragraph one: individuals usually do not recognize relatedness per se (in fact I don't know of any case of strict kin-recognition in social insects), but use some proxy of relatedness (spatial proximity, colony odor, etc.);

We revised this statement as follows: “adaptive organismal social interactions often depend on the ability of individuals to recognize kin and/or group-members to increase cooperation or to suppress inbreeding”

Paragraph three: I'm not sure this really is the consensus in the field. My impression is that genetics does play an important role in this context, but that other factors (like the ones you list below) are important as well. The Gestalt odor then integrates these different factors over the individuals in a colony;

Here we argue that the overall emerging consensus in the literature is that relative to hive and other environmental factors, genetic background seems to play a very minor role in defining the colony-specific nestmate recognition cue in honey bees. In our opinion, the data we present here further support this view. Nevertheless, we toned-down this statement in the revised text as follows:

“However, empirical studies revealed that, surprisingly, colony and social environmental factors play the most dominant role in defining colony-specific cues, and can often mask genetic relatedness (Breed et al., 1988; Downs and Ratnieks, 1999; Heinze et al., 1996; Lahav et al., 2001; Liang and Silverman, 2000; Singer and Espelie, 1996; Stuart, 1988).”

Also in paragraph three: why "passive"? Many of the factors thought to contribute to this process are active (allogrooming, trophalaxis…). For example, queens of social parasites often actively groom and chew on resident workers before entering a host colony. This seems to be an adaptive behavior specifically aimed at achieving chemical camouflage;

Our original reason for using the term “passive” here was to contrast the “passive” acquisition of chemicals from other sources vs. “active” production of the chemicals by the individual. In retrospect, this is clearly confusing. To clarify, we deleted this adverb throughout the revised text.

Again in paragraph three: I'm not aware of ant species that do not display allogrooming behavior (can you provide a reference?);

We thank the reviewers for pointing out the mischaracterization of the phenotype. We revised the text as follows:

“For example, some ant species do not display robust trophallaxis behaviors”

Paragraph four: maybe mention above that the "blank slate" idea also applies to ants?

We revised the text as follows:

“Based on these studies, it has been hypothesized that, similar to some ant species, the CHC profile of newly enclosed workers represents a “blank slate” (Breed et al., 2004; Lenoir et al., 1999), and that, similar to predictions of the classic “Gestalt” model, nestmate recognition cues are subsequently acquired by individual workers via the homogenization and transfer of chemicals through direct social interactions and intermediate environmental factors (Breed et al., 2015).”

3) Figure 1 and related passages: subsection “CHC profiles of individual honey bee workers exhibit qualitative and quantitative age-dependent changes”: There is no significant increase between days 1 and 7. I would reformulate this to "increases between one-day post-reintroduction and 14 days post-reintroduction and then remains stable";

We revised the text as follows:

“This analysis revealed that the total amount of CHCs increases between one-day post-reintroduction and 14-days post-reintroduction and then remains stable”

I would also mention the post-hoc statistics showing which groups exactly are different from each other here. This is currently only given in the figure itself. It's kind of nice though, because the qualitative changes seem to broadly coincide with the quantitative changes in Figure 1A.

We have revised the text as follows:

“(Figure 1B, Permutation MANOVA, F(1,31) = 22.86, R^2^ = 0.43, p < 0.001, FDR pairwise contrasts: Day 1 vs. Day 7 p = 0.002, Day 1 vs. Day 14 p = 0.002, Day 1 vs. Day 21 p = 0.002, Day 7 vs. Day 14 = 0.017, Day 7 vs. Day 21 p = 0.002, Day 14 vs. Day 21 p = 0.31; Figure 1 —figure supplement 1B, Permutation MANOVA, F(3,28) = 2.35, R^2^ = 0.22, p = 0.038, FDR pairwise contrasts: Day 1 vs. Day 7 p = 0.024, Day 1 vs. Day 14 p = 0.011, Day 1 vs. Day 18 p = 0.018, Day 7 vs. Day 14 = 0.406, Day 7 vs. Day 18 p = 0.212, Day 14 vs. Day 18 p = 0.524)”

Regarding this analysis and the data in Table 1, is there a technical reason that limits you to working with proportions? The way you analyze these data forces you to keep the change in overall amount separate from "qualitative", i.e. relative changes in proportion. But it seems that the picture would be much clearer if you could simply show which compounds change quantitatively with age, and which ones do not. As it is now, you might well recover compounds that change in relative proportion, but that are completely invariable in terms of quantity, while you might miss some that change significantly in quantity but remain relatively stable in overall proportion. It's straightforward to calculate proportions from the quantities of all compounds, but you can't do it the other way around. So I feel that presenting the data the former way would be more informative. It would also be helpful to use the same color codes for different compounds in the Figure 1C,D and Figure 1—figure supplement 1C,D panels.

Since most studies in this field rely on relative proportions of individual compounds and overall profiles using relative proportions, we believe that it is helpful to show differences in relative proportions between bees of different ages. However, we agree with the reviewers that changes in relative proportion do not necessarily depict changes in quantity, so we now also include graphs and tables depicting changes in quantity as follows:

Changes in quantity: Table 1, Table 2, Figure 1C,D, Figure 1—figure supplement 1C,D.

Changes in proportion: Table 3, Table 4, Figure 1E,F, Figure 1—figure supplement 1E,F.

4) Figure 2 and related passages: The environmental experience of undertakers and foragers is probably quite different, in contrast to the perspective of the authors. Unless the authors can determine the time spent outside for undertakes and foragers, it is plausible that differences in environmental exposure contribute to CHC profiles, putting a hole in the argument that environmental exposure is not important to the development of CHC profiles. This is something that the authors will have to grapple with in their revision.

We now include new data that addresses this comment. We have made the following modifications to the results and conclusions of this section:

“These data suggest that some outdoor exposure is not sufficient to drive forager-specific CHC signatures. We next asked whether the CHC signatures of foragers are a direct consequence of their behavioral state by using “big back colonies” (Ben-Shahar et al., 2000; Withers et al., 1995), which allowed us to compare active foragers to bees of a similar age and behavioral state that are unable to forage outside (see Materials and methods). We found that the overall CHC profiles of “big-back” bees were different from those of their actively foraging sisters (Figure 2E, Permutation MANOVA, F(1,14)=5.91, R^2^ = 0.313, p < 0.001). These data suggest that the physiological transition to foraging behaviors is not the sole factor that defines forager-specific CHC profiles, and that it could be modulated by additional factors associated with the act of foraging itself and/or extended exposure to various outdoor environmental factors. However, the fact that foraging nestmates express very similar CHC profiles, which are markedly different from those of non-nestmate foragers sharing a similar foraging environment (Figure 2—figure supplement 1, Permutation MANOVA, F(1,15) = 12.5, R^2^ = 0.47, p < 0.001) suggests that forager-specific chemical signatures are not simply defined by the foraging environment. Additionally, to test whether extended exposure to outdoor environmental factors induces predictable changes in CHCs, we compared the relative amount of individual compounds between forager bees and in-hive bees across our various experiments. We did not find a single compound that varied between foragers and in-hive bees in a consistent manner across our experiments (e.g. always increases or always decreases in association with foraging activity) (Table 5), indicating that CHCs do not change in a stereotypic manner in association with extended outdoor exposure, as they do in Harvester ants (Wagner et al., 2001). Nevertheless, to further examine whether forager-specific CHC profiles were solely environmentally determined, we also analyzed the CHC profiles of typical-age foragers that were forced to revert to a nursing state (Robinson et al., 1992). However, we did not find any differences between the CHC profiles of reverted nurses and active foragers (Figure 2F). These data suggest that once foragers acquire their signature CHC profile, it remains stable independent of the task they perform and despite the typical short CHC half-life in insects (Kent et al., 2007). Together, these data suggest that forager-specific CHC signatures are derived from a combination of factors associated with an innate behavioral maturation process, as well as being physically engaged in foraging activity.”

Additionally, in subsection “The CHC signatures of individual workers are task-related”: I find it confusing that you present significant p-values to back up a statement that something is not different.

We agree that this is confusing. We have now included the post-hoc statistics here to demonstrate that foragers and nurses are not statistically significantly different. The statistics originally cited were for the overall MANOVA, which included nurses, undertakers and foragers, and whose low p-value was driven by comparison of foragers with nurses.

“(Figure 2D, Permutation MANOVA, F(2,23)=12.60, R^2^ = 0.55, p < 0.001, FDR pairwise contrasts: undertaker vs. forager p = 0.003, undertaker vs. nurse p = 0.176, forager vs. nurse p = 0.003)”

Also, how old are the undertakers? According to what you write above, they should differ as long as undertakers are less than seven days and foragers are more than seven days old.

Undertakers are usually 2-3 weeks old, which we have made note of in the text.

“’Undertakers’ are a small group of highly specialized older pre-foraging workers (2-3 weeks of age), which are responsible for removing dead bees by carrying them outside and away from the colony (Robinson, 1992; Smith et al., 2008; Søvik et al., 2015; Trumbo et al., 1997).”

If you look at Figure 2D it indeed looks like the undertakers have less CHCs than foragers, even though the difference is not statistically significant. I would therefore be careful with how you phrase this here. Something like "we did not detect a statistically significant difference" might be better than "we found that.… undertakers do not differ from foragers". Such a negative finding would have to be backed up by a careful statistical power analysis and then refer to specific effect sizes etc. So to me it looks like undertakers are very similar to nurses both in terms of total amount and profile quality; Second paragraph, final sentence: Is this a valid comparison?

We have included new data that addresses this comment, as described in our response to point 4.

5) Figure 3 and related passages: subsection “The development of individual CHC profiles is a regulated process modulated by the colony environment”: Here again, I think this should be formulated more carefully. This section hinges on the conclusion that workers from the same colony don't differ at days 7 and 14, but they do at foraging onset. So for the important negative results the authors should present a careful power analysis to show that they would be able to detect the kinds of effect sizes observed between foragers if they were present at earlier time points. Looking at Figure 3, this is really far from clear – the data actually look quite messy in that there appears to be some separation in all three cases, but there is also always a lot of overlap. So you'd really have to convince the reader that C is in fact different from A and B.

To address this comment, we now include a multivariate power analysis as described in (Anderson and Santana-Garcon, 2015) in Figure 3—figure supplement 1. Additionally, while our visualization method is separate from our statistical test, it unfortunately does not fully recapitulate the statistical significance found in this data set, therefore we now include more detailed statistics as follows:

“CHC analyses revealed that through Day 14, the CHC profiles of bees were more similar to the profiles of their same-aged non-nestmate sisters than those of unrelated nestmates of similar age (Figure 3A, Two-way Permutation MANOVA, foster colony (environment): F(1,31) = 2.19, R^2^ = 0.06, p = 0.06, source colony (genetics): F(1,31) = 5.94, R^2^ = 0.16, p < 0.001, foster colony*source colony: F(1,31)=0.46, R^2^ = 0.01, p = 0.82; Figure 3B, Two-way Permutation MANOVA, foster colony: F(1,31) = 1.13, R^2^ = 0.03, p = 0.33, source colony: F(1,31) = 3.18, R^2^ = 0.09, p = 0.02, foster colony*source colony: F(1,31) = 1.78, R^2^ = 0.05, p = 0.15; sample size assessment depicted in Figure 3—figure supplement 1 indicates sample size is adequate).”

I also find the term "natal sisters" confusing. If I understand correctly, these are the sisters that were put back into the natal colony of the fostered individuals, but they are not natal to the colony the experimental bees were fostered into. I would reformulate this to avoid the possible confusion.

We thank the reviewers for pointing out this confusion. We have eliminated this term and have clarified these comparisons in the text.

Change "factors associated with the natal colony" to "genetics or other factors associated with the natal colony"?

We thank the reviewers for this suggestion and have made this change:

“Together, these data suggest that genetic variations, or other long-term effects associated with the source colony, play an important role in defining the CHC profiles of individuals during the early phases of the age-dependent behavioral development of worker honey bees.”

6) Figure 4: This figure is not very informative and I would move it to the supplement. The trees are essentially not resolved – I would collapse all nodes with less than 50 percent bootstrap support – the displayed topology is quite misleading. Also, for ML analyses it's more informative to display the single ML tree with branch lengths and then map bootstrap support onto that tree (rather than presenting a consensus tree without branch length). Finally, how were the outgroups chosen and how can you be sure they're actual outgroups? I couldn't find any justification in the Materials and methods.

To address this concern, we replaced the protein tree with a table that describes the protein BLAST E-values used to identify honey bee genes that encode enzymes that play a role in insect CHC biosynthesis by using well-characterized representative genes in *Drosophila melanogaster* (Table 6).

7) Figure 5 and related passages: The authors chose to conduct qPCR to measure gene expression, and they use a single reference gene based on previously published studies. The standard for these types of analyses would be to use several reference genes that are first validated in the specific context of a given study.

We agree with the reviewers that a careful choice of control genes is essential for robust quantification of mRNA expression levels by using relative qRT-PCR data. We also agree that in some experimental contexts, using more than one control gene increases the confidence that the described changes in gene expression represent differences in the focal rather than changes in the expression of the putative “control” gene. In fact, in the current study we originally attempted to use the same two control genes we (Greenberg et al., 2012) and others have used in past honey bee studies. However, in contrast to our previous studies of brain tissues, here we found that only the *EIF3-S8* gene was reliably amplified from abdominal RNA tissues. In contrast, the same *Arp1* assay we successfully used in brain studies does not work reliably with abdominal tissues. Therefore, after careful consideration, we decided to only use a single control gene for C_t_ normalization instead of the geometric average of two different control genes. Although previously published studies by others have used *EIF3-S8* as a single control gene expression in studies of gene expression in honey bee cuticular tissues (e.g. Richards et. al, 2008), we made sure that the raw C_t_ values of *EIF3-S8* were not affected by any of the experimental conditions included in the current studies. Nevertheless, to clarify this important technical point, we added the following statement:

“Expression levels of the *EIF3-S8,* a housekeeping gene that has previously been used as a reference gene in honey bee studies of gene expression by us and others (Alaux et al., 2009; Fischer and Grozinger, 2008; Greenberg et al., 2012; Mao et al., 2015; Richard et al., 2008), was used as a loading control. To further ensure that the reported expression data for the experimental genes are robust, we first confirmed that the raw *EIF3-S8* C_t_ values per total RNA used in the individual RT reactions were not affected by any of the studied groups included in our current study (Kruskal-Wallis, H = 3.299, df = 4, p = 0.5091).”

Furthermore, there are quite a few genes being tested, and that should be taken into account in multiple-comparison corrections, i.e. p-values for significance cutoffs should be adjusted accordingly. Based on the current data I could probably be convinced that there is some dynamic gene regulation going on, but looking at Figure 5 and the associated supplementary table, I find it really hard to say which of the indicated changes are actually real and biologically meaningful and which ones are spurious.

To address this concern, we added FDR adjusted p-values to Table 7, and now only show expression data for genes whose FDR adjusted p-value is < 0.05 (Figure 5).

Also, given the availability of RNA-seq approaches, it seems surprising that the authors still opted for qPCRs – RNA-seq would be much broader in scope and the statistical analysis would be more robust. I therefore feel that this is somewhat of a weakness of this study, and if at all possible, I would collect RNA-seq data in addition to the qPCR data. Patterns that hold up across both datasets would then be quite convincing.

We respectfully disagree with the reviewers that not using RNA-seq data represent a weakness of our study. In our experience, RNA-seq is an excellent hypothesis-generating approach, which we often use as an effective tool for the discovery phase of our projects. However, we find that in our hands, the unavoidable high level of quantitative noise, and the obvious statistical issues associated with multiple comparisons, significantly reduce the detection power of RNA-seq, especially for small-effect genes, which makes it a weak tool for testing gene-centric hypotheses. However, because here we are testing a very specific hypothesis about a group of candidate genes, we hope that the reviewers would agree with us that the actual potential for gaining additional mechanistic insights is minimal, and therefore, does not justify the additional costs and time associated with adding the suggested RNA-seq study.

Figure 5 and the legend: remove indications of p<0.1 – these are not significant.

The revised text only describes significant p-values (p<0.05).

8) The authors present data showing that CHCs differ across ages and tasks and, separately, that expression of elongases and desaturases differ across ages and tasks, and the authors argue that these are manifestations of the same underlying process. Thus, it seems important to test the relationship between these two. For example, since desaturases catalyze the formation of double bonds in alkenes, are there more alkenes produced by bees that express higher levels of desaturases?

The bees used for the qPCR analysis are from the same group(s) as those in Figure 1—figure supplement 1, which demonstrate this relationship. We have made a note of this in the manuscript (“source colony (sisters of the bees analyzed for Figure 1—figure supplement 1)”). Unfortunately, bees are unable to be used for both CHC extraction and RNA extraction (CHC extraction occurs at room temperature and likely degrades RNA, RNA extraction requires dissection, which makes the animal unusable for CHC extraction), so the exact same bees cannot be used to identify an exact relationship.

In terms of linking the two analyses, this would certainly be an interesting future direction, but something that we cannot currently address. Because these enzymes likely have complex function, are involved in other processes other than producing CHCs, might have varying threshold values of when they are active, etc., it is difficult to make claims about specific enzymes controlling specific CHCs. Indeed, we do see an increase in some desaturase genes and some alkenes, but because the relationships between specific enzymes and compounds are complex, we do not feel comfortable to speculate about any possible causal relationships. Therefore, to avoid overinterpretation of our data, we currently focus our data interpretations on the intrinsic changes in gene expression levels in association with age and hive environment to support the more conservative data interpretation that age and the hive environment influence innate processes related to CHC synthesis. We address this in the revised text as follows:

“[These data] indicate that the overall observed qualitative age- and task-dependent changes in the CHC profiles of individual workers are affected by intrinsic molecular dynamics of the CHC synthesis pathway. However, our studies also importantly show that genetically-related bees that age in different colonies exhibit qualitatively different CHC profiles and CHC biosynthesis gene expression levels, which suggests that the CHC synthesis process is also plastic, and could be modulated by factors associated with the hive/ social environment.”

9) Figure 6 and related passages: subsection “Age and task play a role in defining nestmate recognition cues in honey bee colonies.”: Shouldn't statistical comparisons be between natal foragers and other types of natal bees? It's not intuitive to say that they are accepted by showing comparisons that show significant differences. Figure 6A: Again, this is not very intuitive. What this panel is supposed to show is that bees of all ages are accepted into their own colony at the same rate as foragers from the same colony are. However, all the statistical comparisons are between natal bees and bees from a different colony, rather than between different types of natal bees and natal foragers. What this really shows is that, as expected, natal bees of any kind are accepted into their colony more readily than foreign foragers. If that's the message of this figure then I would present it as such. If, on the other hand, this is supposed to compare different types of natal bees to natal foragers as a baseline, then the statistics would have to be done differently. For example, it looks like 14 day old bees are much more readily accepted into the colony than foragers. If these statistical comparisons have in fact been done and they are all non-significant, then I would indicate this with bars labeled "ns" in the figure. It would also be helpful to have standard errors on the bars. First paragraph, penultimate sentence: Same problem here. All of these comparisons are statistically significant, so it's not obvious how they support the claim made here. These don't seem to be the most relevant statistical comparisons. Figure 6B: Similar problems here. Comparisons that are not significant should be indicated.

We thank the reviewers for pointing out this somewhat confusing data presentation. To clarify, we now describe all possible pair-wise comparisons, including those that are not significant (Figure 5). Because these comparisons are based on Pearson’s Chi-square analyses of proportional data (accepted/rejected bees), standard error bars cannot be applied.

Also in paragraph two: I'm not sure how this follows from the previous experiments and the last experiment (Figure 6C) doesn't seem particularly relevant to the rest of the study. The alarm pheromone experiment is also challenging as different doses may yield different results. Removing Figure 6C would strengthen the manuscript.

We agree that this figure does not fit with the main findings of the paper, and have decided to remove it from the revised manuscript.

[Editors' note: further revisions were requested prior to acceptance, as described below.]

The manuscript has been improved but there are a few textual changes that need to be addressed before acceptance. Specifically, care should be taken to use the term CHCs instead of chemical signatures or recognition cues, as detailed below. The discussion of the Gestalt model and implications from this study should be more nuanced, as described below.Reviewer #2:The revised manuscript is much improved, especially the way the results are presented. There are still a couple of issues where I think the authors should present their findings more carefully though. Maybe most importantly, as was already noted in the first round of reviews, it is not known which cues honeybees actually use for non-nestmate discrimination. What the authors study here are the dynamic changes in cuticular hydrocarbon profiles, without knowledge of their actual biological function. It is of course reasonable to say in the Introduction that CHCs have been implicated in non-nestmate discrimination, and to later discuss the possible implications of their findings in that context, but I would leave it at that, and be very careful in how to phrase this. I think that would further strengthen the manuscript. For example, I would change the title from "Chemical signatures of honey bee group membership…" to "Honey bee cuticular hydrocarbon profiles…" – you really don't know whether the observed changes are signatures of group membership, at least in any biologically meaningful sense. If anything, the results in subsection “The development of individual CHC profiles is a regulated process modulated by the colony environment”, for example, suggest that they are not. Similarly, in the Abstract, I would replace all mentions of "nestmate recognition cues" with "cuticular hydrocarbon profiles". Same in Discussion paragraph one. Introduction final paragraph: you don't know what the "colony-specific nestmate recognition cues" are, so you can't say anything about that directly.

We agree with the reviewer that not all changes in the CHC profiles of individual workers we have identified here are directly relevant to the cues that regulate nestmate recognition. Therefore, we now make it clearer throughout the revised text that the specific identity of the specific compounds that comprise the cue for nestmate recognition remain unknown, and that not all changes in the CHC profile of individuals are directly related to the development of nestmate recognition cues, and place more separation in the conclusions drawn from our CHC results and nestmate recognition cues as described below:

1) We have revised the title as follows: “The cuticular hydrocarbon profiles of honey bee workers develop via a socially-modulated innate process”

2) “Here we show that in the honey bee (*Apis mellifera*), the colony-specific CHC profile completes its maturation in foragers…”

3) “Therefore, the CHC profiles of individual honey bees are not likely produced through homogenization and transfer mechanisms, but instead mature in association with age-dependent division of labor.”

4) “Whether the overall profile, or more specific components of it, represent the actual nestmate recognition cue remains unknown. However, previous studies have indicated that variations in the relative amounts of each compound in the CHC profile across individuals from different colonies are likely sufficient for the chemical recognition of nest membership”

5) “Here we provide empirical evidence that the maturation of the CHC profile of individual honey bee workers is primarily regulated by innate developmental processes associated with age-dependent behavioral tasks and modulated by the social colony environment, and that mature colony-specific recognition cues are primarily associated with the foraging task.”

6) “Instead, our data indicate that CHC profiles, and likely nestmate recognition cues, in honey bees are more likely a product of a genetically-determined developmental program that is modulated by colony-specific factors.”

7) “However, the data we present here suggest that the overall development of individual CHC profiles of honey bee workers primarily depends on an innate developmental process”

8) “Therefore, we posit that it is unlikely that CHC profiles in honey bees develop through homogenization and transfer mechanisms between nestmates and hive materials. Furthermore, given the established implicated role of CHCs in nestmate recognition (van Zweden and D’Ettorre, 2010), we additionally posit that CHC homogenization mechanisms are unlikely to play a key role in the production of colony-specific cues in honey bees.”

9) “Which specific components of the honey bee CHC profile represent the nestmate recognition cue remains unknown. Although it has been shown that CHCs are likely used for nestmate recognition in honey bees (van Zweden and D’Ettorre, 2010), it is unlikely that all components of the CHC profile contribute to this process (Akino et al., 2004; Dani et al., 2005, 2001; Martin et al., 2008; Ruther et al., 2002).”

As an aside, and I know it sounds bulky, I prefer "non-nestmate discrimination" over "nestmate recognition". At a behavioral level, what is usually measured is that social insects discriminate against (attack) non-nestmates, not whether they recognize nestmates (you could make a similar argument for models of the underlying neuronal computation). I know that's not necessarily reflected in the literature and I would therefore leave it up to the authors, but in my opinion this is a better way to think about the behavior.

We thank the reviewer for bringing up this important distinction. In our view, describing the main focus of this study as either "non-nestmate discrimination" or "nestmate recognition" is contextual. We prefer to continue using the term "nestmate recognition" when referring to this trait in general, and "non-nestmate discrimination" when referring to the actual behavioral assays used to measure it. We clarify this in the revised manuscript as follows:

“One remarkable example of organismal recognition of “self” comes from colonies of social insects, which depend on a robust non-nestmate discrimination system (more commonly called “nestmate recognition”) to prevent the loss of expensive resources to non-nestmates, and to maintain overall colony integrity (Hefetz, 2007; van Zweden and D’Ettorre, 2010).”

I still find it somewhat unfortunate how the study is set up in the Introduction, and how the findings are summarized. E.g. "Yet, whether "Gestalt"-like mechanisms regulate nestmate recognition systems in other social insect species remains mostly unexplored."; "However, whether "Gestalt"-like homogenization mechanisms play a role in honey bee nestmate recognition remains unknown."; and then the brief summary at the start of the final paragraph of the Introduction, saying that the authors' findings from honey bees contradict a Gestalt model. I don't think they do. You really don't know what the nestmate recognition cues are, and whether you're even capturing them in your analyses, so you can't exclude that the Gestalt model actually applies to these cues, and that the age-dependent changes you pick up here are either irrelevant (to the bees), or represent e.g. desiccation barriers and/or task group recognition cues that are superimposed on colony recognition cues (but have nothing to do with non-nestmate discrimination). Same problem Discussion final paragraph. Subsection “The CHC signatures of individual workers are task-related” paragraph three, sentence four: Doesn't this argue for some sort of Gestalt (or maybe genetic) effect?

We regret not making our rationale and data interpretations clearer. The primary hypothesis we tested here was derived from the assumptions originally proposed by the gestalt model, which suggested that colony-specific chemical cues are an emergent phenomenon that develops via CHC homogenization and transfer across nestmates. Previous studies suggested that in the honey bees, this mechanism may explain the development and propagation of colony specific chemical signatures. However, the data we present here suggest that this process in more complex than originally thought, and involves both innate processes as well as interactions with colony factors. Nevertheless, to address this concern, we have toned down our data interpretations in terms of its implication for the overall validity of the gestalt model. Therefore, we removed the term “gestalt” from most of the manuscript, and refocused it on determining when and how CHC profiles and recognition cues develop in honey bees, and whether CHC homogenization mechanisms might be sufficient to explain our observation. Key revised statements are described below:

1) “Consequently, here we investigated the development of CHC profiles and nestmate recognition cues in the European honey bee, *Apis mellifera*”

2) “However, when and how honey bee chemical nestmate recognition cues mature, and whether CHC homogenization mechanisms play a role in this process have not been directly investigated.”

3) “Therefore, we posit that it is unlikely that CHC profiles in honey bees develop through homogenization and transfer mechanisms between nestmates and hive materials. Furthermore, given the established implicated role of CHCs in nestmate recognition (van Zweden and D’Ettorre, 2010), we additionally posit that CHC homogenization mechanisms are unlikely to play a key role in the production of colony-specific cues in honey bees.”

4) “Therefore, our data suggest that CHC profiles of honey bee workers develop via a biphasic process…”

5) “In conclusion, we propose that nestmate recognition cue production and acquisition in honey bees are not likely to be primarily driven by CHC homogenization and transfer mechanisms as previously described in some ant species”